A striking new genus and species of cave-dwelling frog (Amphibia: Anura: Microhylidae: Asterophryinae) from Thailand

Suwannapoom Chatmongkon 1
Sumontha Montri 2 montri.sumontha@gmail.com
Tunprasert Jitthep 3
Ruangsuwan Thiti 4
Pawangkhanant Parinya 1
Korost Dmitriy V. 5
http://orcid.org/0000-0002-7576-2283 Poyarkov Nikolay A. 6 7 n.poyarkov@gmail.com
1 Division of Fishery, School of Agriculture and Natural Resources, University of Phayao , Phayao , Thailand
2 Department of Fishery, Ranong Marine Fisheries Station , Ranong , Thailand
3 Department of Ecology, Nakhon Pathom Rajabhat University , Nakhon Pathom Mueng, Nakhon Pathom , Thailand
4 Department of Zoology, Faculty of Science, Kasetsart University , Bangkok , Thailand
5 Geological Faculty, Petroleum Geology Department, Moscow State University , Moscow , Russia
6 Biological Faculty, Department of Vertebrate Zoology, Moscow State University , Moscow , Russia
7 Laboratory of Tropical Ecology, Joint Russian–Vietnamese Tropical Research and Technological Center , Hanoi , Vietnam
Parra Olea Gabriela
Electronic publication date: 2018 Feb 23
Publication date: 2018
Volume: 6
Electronic Location ID: e4422
Received 2017 Dec 21; Accepted 2018 Feb 7
Copyright: © 2018 Suwannapoom et al.
Copyright year: 2018
Copyright holder: Suwannapoom et al.
License: This is an open access article distributed under the terms of the Creative Commons Attribution License, which permits unrestricted use, distribution, reproduction and adaptation in any medium and for any purpose provided that it is properly attributed. For attribution, the original author(s), title, publication source (PeerJ) and either DOI or URL of the article must be cited.
License URL: https://creativecommons.org/licenses/by/4.0/

Keywords: Kanchanaburi Province, Siamophryne troglodytesGen. et sp. nov., Tadpole, Troglophilous life style, Tenasserim, Sundaland, mtDNA, Biogeography, microCT-scanning

Funding: Thailand Research Fund DBG6180001 Russian Foundation for Basic Research RFBR 15-29-02771 Russian Science Foundation 14-50-00029 This work was supported by the Thailand Research Fund (DBG6180001), and by the Russian Foundation for Basic Research (Grant No. RFBR 15-29-02771) (fieldwork, specimen examination). Molecular experiments, phylogenetic analyses, specimen storage, examination, and microCT-analysis were carried out with the financial support of Russian Science Foundation (RSF grant No. 14-50-00029). The funders had no role in the study design, data collection and analysis, decision to publish or preparation of the manuscript.

==============================
We report on a discovery of Siamophryne troglodytes Gen. et sp. nov., a new troglophilous genus and species of microhylid frog from a limestone cave in the tropical forests of western Thailand. To assess its phylogenetic relationships we studied the 12S rRNA–16S rRNA mtDNA fragment with final alignment comprising up to 2,591 bp for 56 microhylid species. Morphological characterization of the new genus is based on examination of external morphology and analysis of osteological characteristics using microCT-scanning. Phylogenetic analyses place the new genus into the mainly Australasian subfamily Asterophryinae as a sister taxon to the genus Gastrophrynoides, the only member of the subfamily known from Sundaland. The new genus markedly differs from all other Asterophryinae members by a number of diagnostic morphological characters and demonstrates significant mtDNA sequence divergence. We provide a preliminary description of a tadpole of the new genus. Thus, it represents the only asterophryine taxon with documented free-living larval stage and troglophilous life style. Our work demonstrates that S. troglodytes Gen. et sp. nov. represents an old lineage of the initial radiation of Asterophryinae which took place in the mainland Southeast Asia. Our results strongly support the “out of Indo-Eurasia” biogeographic scenario for this group of frogs. To date, the new frog is only known from a single limestone cave system in Sai Yok District of Kanchanaburi Province of Thailand; its habitat is affected by illegal bat guano mining and other human activities. As such, S. troglodytes Gen. et sp. nov. is likely to be at high risk of habitat loss. Considering high ecological specialization and a small known range of the new taxon, we propose a IUCN Red List status of endangered for it.

Introduction

Microhylidae is one of the largest frog families belonging to the Ranoidea, with a pan-tropical distribution. To date, it includes 641 species (nearly 9.4% of anuran diversity) (Frost, 2017). Microhylid frogs occur on most of the continents and several large islands, their range encompasses tropical and subtropical areas of Southern and Northern America, Africa, Madagascar, South, Southeast, and East Asia, Australasian islands, and northern Australia. Family-level taxonomy of Microhylidae is considered to be a “phylogeneticist’s nightmare” (Peloso et al., 2016) and is the subject of numerous studies and an ongoing debate. At present, 13 subfamilies are recognized based on morphological and molecular phylogenetic data (Matsui et al., 2011; Pyron & Wiens, 2011; Kurabayashi et al., 2011; de Sá et al., 2012; Peloso et al., 2016). However, the degree of coherence between the morphological and molecular classifications of the family is quite low due to the high morphological variation and widespread convergence in microhylids, which, in many cases, likely is connected to specializations associated with a burrowing lifestyle (see de Sá et al., 2012; Peloso et al., 2016). The basal split within Microhylidae is estimated to coincide with the Cretaceous–Paleogene boundary (65 Ma) (Feng et al., 2017); previous studies argued for Mesozoic origin of Microhylidae and considered the constituent subfamilies as full families of Anura (Bossuyt & Roelants, 2009).

Each Microhylidae subfamily is restricted to a landmass derived from the breaking up of the Gondwana: the Americas (subfamilies Adelastinae, Gastrophryninae, and Otophryninae), Africa (subfamilies Hoplophryninae and Phrynomerinae), Madagascar (subfamilies Cophylinae, Dyscophinae, and Scaphiophryninae), India (subfamily Melanobatrachinae), East, South, and Southeast Asia (subfamilies Chaperininae, Kalophryninae, and Microhylinae), and Australasia (subfamily Asterophryinae) (Kurabayashi et al., 2011; de Sá et al., 2012; Peloso et al., 2016). Due to their transcontinental pantropical distribution, Microhylidae were regarded as a promising model group for biogeography studies (Savage, 1973). Most previous works, though varying on taxon sampling and molecular data, suggested that Microhylidae are of Gondwanan origin and gave evidence supporting the “Antarctic route scenario” for the Australasian subfamily Asterophryinae, as suggested for several other vertebrate taxa that are distributed in Australia (Van Bocxlaer et al., 2006; Van der Meijden et al., 2007). According to this scenario, the basal split of Microhylidae took place in Gondwana and the ancestor of Asterophryinae dispersed to Australia via Antarctic land bridge (Hill, 2009), where the subfamily diversified (it comprises 323 recognized species to date, Frost, 2017) and subsequently dispersed to New Guinea and adjacent Australasian islands, but was unable to cross the Wallace line with exception of the genus Oreophryne Boettger, which is also known from the island of Bali (west from the Wallace line, see Fig. 1).

Figure 1 Known distribution of main Asterophryinae lineages and the type locality (green star) of Siamophryne troglodytes Gen. et sp. nov. in Kanchanaburi Province, Thailand.

Biogeographic borders: (A) the Isthmus of Kra line, the approximate biogeographic border between Sundaland and Indochina; (B-1) the Wallace line (after Huxley, 1868); (B-2) the Wallace line (after Mayr, 1944); (C) the Weber line; (D) the Lyddeker line. The majority of the Asterophryinae genera inhabit Australasia east of the Wallace line (red) and the island of Bali; Gastrophrynoides is confined to Sundaland (Borneo and Peninsular Malaysia); Siamophryne troglodytes Gen. et sp. nov. is known from Indochina.

However, for the first time, Matsui et al. (2011) reported on the phylogenetic position of the enigmatic genus Gastrophrynoides Noble inhabiting Sundaland (Peninsular Malaysia and northern Borneo, see Fig. 1), which was not yet assigned to any certain subfamily. Based on the analysis of 16S rRNA–12S rRNA mtDNA data it was established as a sister lineage of the genus Oreophryne (Asterophryinae). Association of Gastrophrynoides with asterophryines was further suggested by Kurabayashi et al. (2011), what allowed the authors to assign Gastrophrynoides to the subfamily Asterophryinae and propose an alternative biogeographic scenario for the group. According to Kurabayashi et al. (2011), Gastrophrynoides separated from other asterophryines around 48 Ma, while the presence of the most basal asterophryine taxon in the Eurasian area (Sundaland) suggests that the colonization route of Asterophryinae goes from Asia to Australia, but not via Antarctica as was suggested earlier. Further studies, applying multilocus (de Sá et al., 2012) and phylogenomic (Peloso et al., 2016) approaches, strongly supported the placement of the Sundanese Gastrophrynoides as a sister group to all other genera of the subfamily inhabiting Australasia.

As Kurabayashi et al. (2011: 9) stated: “the biogeographic findings on Gastrophrynoides imply the possible occurrence of further microhylid taxa with unexpected evolutionary backgrounds and give a basis for future paleontological and biogeographic studies of Asian anurans.” In 2016, during a field survey in a limestone cave system in Kanchanaburi Province of western Thailand, we encountered an unusually-looking troglophilous frog. It was assigned to the family Microhylidae due to the presence of the following traits: lack of maxillary teeth, lack of parotoid glands, and a firmisternal pectoral girdle with nonoverlapping epicoracoids, well developed coracoids reaching the midline of the girdle and scapulae, a large, cartilaginous sternum, reduced clavicles and no omosternum. However, the new frog was morphologically distinct from any genus of Microhylidae known to occur in Thailand or adjacent parts of Indochina. Further detailed morphological, osteological, and phylogenetic analyses indicated that the Microhylidae Gen. sp. from Kanchanaburi represents a new yet undescribed genus of Asterophryinae frogs, a sister taxon to Gastrophrynoides. We provide the description of this new frog herein. As we demonstrate below, our discovery carries important biogeographic implications: highlights the initial radiation of Asterophryinae, which took place in the mainland Southeast Asia, and supports the “out of Indo-Eurasia” biogeographic scenario for this group of frogs.

Materials and Methods

Sample collection

Field work was conducted from August to October of 2016 in Sai Yok District, Kanchanaburi Province, northern Tenasserim Region, western Thailand (approximate geographic coordinates: 14.476°N, 98.853°E; elevation—440 m a.s.l.). Geographic coordinates and elevation were obtained using a Garmin GPSMAP 60CSx and recorded in the WGS 84 datum. In total, 11 adult specimens (six males and five gravid females) and a single tadpole of a new microhylid frog were collected and photographed in life before being euthanized using 20% solution of benzocaine prior to fixation in 96% ethanol and were subsequently stored in 70% ethanol. The larval specimen was fixed and subsequently stored in 4% formalin. Tissue samples for genetic analysis were taken prior to preservation and were stored in 95% ethanol. Specimens and tissues were subsequently deposited in the herpetological collections of the School of Agriculture and Natural Resources, University of Phayao (AUP, Phayao, Thailand) and of the Zoological Museum of Moscow University (ZMMU, Moscow, Russia).

Specimens collection protocols and animal use were approved by the Institutional Ethical Committee of Animal Experimentation of the University of Phayao, Phayao, Thailand (certificate number UP-AE59-01-04-0022 issued to Chatmongkon Suwannapoom) and strictly complied with the ethical conditions by the Thailand Animal Welfare Act. Field work, including collection of animals in the field and specimen exportation was authorized by the Institute of Animals for Scientific Purpose Development, Bangkok, Thailand (permit number U1-01205-2558, issued to Chatmongkon Suwannapoom).

The electronic version of this article in portable document format will represent a published work according to the International Commission on Zoological Nomenclature (ICZN), and hence the new names contained in the electronic version are effectively published under that Code from the electronic edition alone (see Articles 8.5–8.6 of the Code). This published work and the nomenclatural acts it contains have been registered in ZooBank, the online registration system for the ICZN. The ZooBank LSIDs (Life Science Identifiers) can be resolved and the associated information viewed through any standard web browser by appending the LSID to the prefix http://zoobank.org/. The LSID for this publication is: urn:lsid:zoobank.org:pub:C8BD1C1D-0553-4662-8DE5-4337CD69E3B9. The online version of this work is archived and available from the following digital repositories: PeerJ, PubMed Central, and CLOCKSS.

Laboratory methods

For molecular phylogenetic analysis, total genomic DNA was extracted from ethanol-preserved femoral muscle tissue using standard phenol–chloroform–proteinase K (final concentration 1 mg/ml) extraction procedures with subsequent isopropanol precipitation (protocols in accordance with Hillis et al., 1996 and Sambrook & David, 2001). The isolated full-genome DNA was visualized using agarose electrophoresis in presence of ethidium bromide. The total concentration of DNA in 1 μl was measured using NanoDrop 2000 (Thermo Scientific, Waltham, MA, USA), and consequently adjusted to ca. 100 ng DNA/μl.

We amplified mtDNA fragments, covering partial sequences of 12S rRNA and 16S rRNA mtDNA genes and a complete sequence of tRNAVal mtDNA gene in order to obtain a 2,591 bp-length continuous fragment of mtDNA. 16S rRNA is a molecular marker widely applied for biodiversity surveys in amphibians (Vences et al., 2005a, 2005b; Vieites et al., 2009). Together with 12S rRNA partial sequences these mtDNA markers were used in the most comprehensive phylogenetic studies on Microhylinae frogs published to date (Matsui et al., 2011; Pyron & Wiens, 2011; de Sá et al., 2012; Peloso et al., 2016, and references therein), including the molecular taxonomic research on the subfamily Asterophryinae (Hoskin, 2004; Frost et al., 2006; Köhler & Günther, 2008; Günther, Stelbrink & von Rintelen, 2010; Kurabayashi et al., 2011; Rittmeyer et al., 2012; Blackburn et al., 2013; Oliver et al., 2013, 2017b). Amplification was performed in 20 μl reactions using ca. 50 ng genomic DNA, 10 nmol of each primer, 15 nmol of each dNTP, 50 nmol of additional MgCl2, Taq PCR buffer (10 mM of Tris–HCl, pH 8.3, 50 mM of KCl, 1.1 mM of MgCl2, and 0.01% gelatine) and 1 U of Taq DNA polymerase. Primers used in PCR and sequencing are summarized in Table 1. The PCR conditions included the following steps: initial denaturation—5 min at 94 °C, 43 cycles of denaturation—1 min at 94 °C, primer annealing—1 min with TouchDown program—reducing the temperature from 65 to 55 °C by 1 °C every cycle, extension—1 min at 72 °C, and final extension—5 min at 72 °C.

Table 1 Primers used in this study.

Primer name	Primer sequence	Reference	
	12S rRNA		
Micro-1F-12stail	ACGCTAAAATGWACCCTAAAAAGT	Nguyen et al. (in press)	
Micro-600R-12stail	TAGAGGAGCCTGTTCTATAATCGATTC	Nguyen et al. (in press)	
Micro-500F-12stail	CCACTTGAACCCACGACAGCTAGRAMACAA	Nguyen et al. (in press)	
Micro-1200R-12stail	AGTAAAGGCGATYAAAAAATRTTTCAAAG	Nguyen et al. (in press)	
12sA-L	AACTGGGATTAGATACCCCACTAT	Palumbi et al. (1991)	
R-1169	GTGGCTGCTTTTAGGCCCACT	Wilkinson, Drewes & Tatum (2002)	
	16S rRNA		
L-2188	AAAGTGGGCCTAAAAGCAGCCA	Matsui et al. (2006)	
16H-1	CTCCGGTCTGAACTCAGATCACGTAGG	Hedges (1994)	

PCR products were visualized using 1.5% agarose electrophoresis in presence of ethidium bromide. If distinct bands were obtained, products were purified prior to cycle sequencing using 2 μl of ExoSapIt (Amersham, Little Chalfont, UK), diluted in the ratio 1:4, per 5 μl of PCR product. A 10 μl sequencing reaction included 2 μl of template, 2.5 μl of sequencing buffer, 0.8 μl of 10 pmol primer, 0.4 μl of BigDye Terminator version 3.1 Sequencing Standard (Applied Biosystems, Waltham, MA, USA), and 4.2 μl of water. The cycle sequencing reaction included 35 cycles consisting of the following steps: 10 s at 96 °C, 10 s at 50 °C, and 4 min at 60 °C. Cycle sequencing products were purified by ethanol precipitation. Sequence data collection and visualization was performed on an ABI 3730xl automated sequencer (Applied Biosystems). The obtained sequences were deposited in the GenBank under the accession numbers MG682553–MG682559 (Table 2).

Table 2 Specimens and sequences of Siamophryne troglodytes Gen. et sp. nov. and outgroup representatives of Microhylidae and Rhacophoridae used in molecular analyses.

Group	GenBank AN	Species	Specimen ID	Reference	
Asterophryinae	DQ283195	Aphantophryne pansa	ABTC 49605	Frost et al. (2006)	
Asterophryinae	FR832625; FR832642	Asterophrys turpicola	ZMB 70537	Günther, Stelbrink & von Rintelen (2010)	
Asterophryinae	JN048979; JN049004	Austrochaperina guttata	LSUMZ 95008	Rittmeyer et al. (2012)	
Asterophryinae	KC822485	Austrochaperina sp.	BSFS 11377	Blackburn et al. (2013)	
Asterophryinae	EU100119; EU100235	Barygenys exsul	BPBM 20128	Köhler & Günther (2008)	
Asterophryinae	KM509105	Callulops robustus	PT-506	Peloso et al. (2016)	
Asterophryinae	DQ283207	Choerophryne sp.	ABTC 47720	Frost et al. (2006)	
Asterophryinae	DQ283206	Cophixalus sphagnicola	ABTC 47881	Frost et al. (2006)	
Asterophryinae	DQ283208	Copiula sp.	AMS R124417	Frost et al. (2006)	
Asterophryinae	AB634647; AB634705	Gastrophrynoides immaculatus	UKMHC 279	Matsui et al. (2011)	
Asterophryinae	DQ283209	Genyophryne thomsoni	ABTC 49624	Frost et al. (2006)	
Asterophryinae	JX119248; JX119392	Hylophorbus rufescens	LSUMZ 94943	Oliver et al. (2013)	
Asterophryinae	DQ283199	Liophryne rhododactyla	ABTC 49566	Frost et al. (2006)	
Asterophryinae	JN048989; JN049014	Mantophryne lateralis	LSUMZ 92102	Rittmeyer et al. (2012)	
Asterophryinae	KM509160	Metamagnusia slateri	PT-507	Peloso et al. (2016)	
Asterophryinae	MG682555	Siamophryne troglodytes Gen. et sp. nov.	AUP-00500	This work	
Asterophryinae	MG682556	Siamophryne troglodytes Gen. et sp. nov.	AUP-00501	This work	
Asterophryinae	MG682557	Siamophryne troglodytes Gen. et sp. nov.	AUP-00502	This work	
Asterophryinae	MG682558	Siamophryne troglodytes Gen. et sp. nov.	AUP-00503	This work	
Asterophryinae	MG682559	Siamophryne troglodytes Gen. et sp. nov. (tadpole)	AUP-00509	This work	
Asterophryinae	MG682553	Siamophryne troglodytes Gen. et sp. nov.	ZMMU A-5818	This work	
Asterophryinae	MG682554	Siamophryne troglodytes Gen. et sp. nov.	ZMMU A-5819	This work	
Asterophryinae	FR832634; FR832635	Oninia senglaubi	ZMB 74608	Günther, Stelbrink & von Rintelen (2010)	
Asterophryinae	KC822488	Oreophryne anulata	PNMCMNHH 1366	Blackburn et al. (2013)	
Asterophryinae	DQ283194	Oreophryne brachypus	ABTC 50081	Frost et al. (2006)	
Asterophryinae	AB634651; AB634709	Oreophryne monticola	MZBAmp 16265	Matsui et al. (2011)	
Asterophryinae	KC822489	Oreophryne variabilis	TNHC 58922	Blackburn et al. (2013)	
Asterophryinae	EU100323; EU100207	Oxydactyla crassa	BPBM 17061	Köhler & Günther (2008)	
Asterophryinae	JN048996; JN049021	Paedophryne amauensis	BPBM 31882	Rittmeyer et al. (2012)	
Asterophryinae	FR832653; FR832636	Pseudocallulops eurydactylus	ZMB 70534	Günther, Stelbrink & von Rintelen (2010)	
Asterophryinae	JX119386; JX119242	Sphenophryne cornuta	LSUMZ 94793	Oliver et al. (2013)	
Asterophryinae	FR832655; FR832638	Xenorhina cf. oxycephala	ZMB 74628	Günther, Stelbrink & von Rintelen (2010)	
Asterophryinae	KM509212	Xenorhina obesa	PT-529	Peloso et al. (2016)	
Chaperininae	AB598318; AB598342	Chaperina fusca	BORN 8478	Matsui et al. (2011)	
Dyscophinae	AB634648; AB634706	Dyscophus guineti	KUHE 33150	Matsui et al. (2011)	
Dyscophinae	AB634649; AB634707	Dyscophus insularis	KUHE 35001	Matsui et al. (2011)	
Gastrophryninae	AB634650; AB634708	Gastrophryne olivacea	KUHE 33224	Matsui et al. (2011)	
Kalophryninae	AB634642; AB634700	Kalophrynus pleurostigma	MZBAmp 15295	Matsui et al. (2011)	
Kalophryninae	AB634645; AB634703	Kalophrynus subterrestris	KUHE 53145	Matsui et al. (2011)	
Melanobatrachinae	KM509159	Melanobatrachus indicus	IND-18	Peloso et al. (2016)	
Microhylinae	AB201182; AB201193	Glyphoglossus molossus	KUHE 35182	Matsui et al. (2011)	
Microhylinae	AB634626; AB634684	Glyphoglossus yunnanensis	KUHE 44148	Matsui et al. (2011)	
Microhylinae	KP682314	Kaloula rugifera	–	Deng et al. (2015)	
Microhylinae	AB634634; AB634692	Metaphrynella pollicaris	KUZ-21655	Matsui et al. (2011)	
Microhylinae	AB634600; AB634658	Microhyla annectens	–	Matsui et al. (2011)	
Microhylinae	DQ512876	Microhyla fissipes	–	Unpublished	
Microhylinae	NC006406	Microhyla heymonsi	–	Zhang et al. (2005)	
Microhylinae	AB303950	Microhyla okinavensis	–	Igawa et al. (2008)	
Microhylinae	AB634616; AB634674	Microhyla petrigena	–	Matsui et al. (2011)	
Microhylinae	NC024547	Microhyla pulchra	–	Wu et al. (2016)	
Microhylinae	AB598317; AB598341	Micryletta inornata	KUHE 20497	Matsui et al. (2011)	
Microhylinae	AB634638; AB634696	Micryletta steinegeri	KUHE 35937	Matsui et al. (2011)	
Microhylinae	AB634636; AB634694	Phrynella pulchra	UKMHC 820	Matsui et al. (2011)	
Microhylinae	AB634633; AB634691	Uperodon taprobanicus	KUHE 37252	Matsui et al. (2011)	
Phrynomerinae	AB634652; AB634710	Phrynomantis bifasciatus	KUHE 33277	Matsui et al. (2011)	
Scaphiophryninae	AB634653; AB634711	Scaphiophryne gottlebei	KUHE 34977	Matsui et al. (2011)	
Rhacophoridae	AB202078	Rhacophorus schlegelii	–	Sano et al. (2005)	
Note:

AN, GenBank accession numbers.

Phylogenetic analyses

For phylogenetic analyses, we used the 12S rRNA and 16S rRNA Microhylidae dataset of Matsui et al. (2011) with addition of available sequences from other Microhylidae genera that are distributed in Southeast Asia and Australasia together with the newly obtained sequences of Microhylidae Gen. sp. from Kanchanaburi Province of Thailand. Data on sequences and specimens used in molecular analyses is summarized in Table 2. In total, sequences of the 12S rRNA and 16S rRNA mtDNA fragments of 56 microhylid representatives and one nonmicrohylid outgroup taxon were subjected to the final analyses, including seven samples of Microhylidae Gen. sp. from Kanchanaburi Province and 44 samples of Asian and Australasian microhylids representing all major lineages of the family inhabiting this region. The subfamily Asterophryinae was represented by approximately 26 species belonging to the following genera: Aphantophryne Fry, Asterophrys Tschudi, Austrochaperina Fry, Barygenys Parker, Callulops Boulenger, Choerophryne Van Kampen, Cophixalus Boettger, Copiula Méhely, Gastrophrynoides, Genyophryne Boulenger (now treated as a junior synonym of Sphenophryne Peters & Doria according to Rivera et al., 2017), Hylophorbus Macleay, Liophryne Boulenger (included in the genus Sphenophryne by Rivera et al., 2017), Mantophryne Boulenger, Metamagnusia Günther (treated as a junior synonym of Asterophrys by Rivera et al., 2017), Oninia Günther, Stelbrink & von Rintelen, Oreophryne, Oxydactyla Van Kampen (considered as synonym of Sphenophryne by Rivera et al., 2017), Paedophryne Kraus, Pseudocallulops Günther (included in the genus Asterophrys by Rivera et al., 2017), Sphenophryne and Xenorhina Peters. Other subfamilies included Microhylinae represented by genera Glyphoglossus Gunther, Kaloula Gray, Metaphrynella Parker, Microhyla Tschudi, Micryletta Dubois, Phrynella Boulenger, and Uperodon Duméril & Bibron (14 species in total), Kalophryninae with a single genus Kalophrynus Tschudi (two species), Melanobatrachinae with a single monotypic genus Melanobatrachus Beddome, and Chaperininae with a single monotypic genus Chaperina Mocquard. Five outgroup sequences of non-Asian Microhylidae included: Dyscophinae (genus Dyscophus Grandidier; two species), Gastrophryninae (genus Gastrophryne Fitzinger; one species), Phrynomerinae (genus Phrynomantis Peters; one species), Scaphiophryninae (genus Scaphiophryne Boulenger; one species) subfamilies. MtDNA sequence of Rhacophorus schlegelii (Günther) (Rhacophoridae; Sano et al., 2005) was used as a nonmicrohylid outgroup.

Nucleotide sequences were initially aligned using ClustalX 1.81 software (Thompson et al., 1997) with default parameters, and then optimized manually in BioEdit 7.0.5.2 (Hall, 1999) and MEGA 6.0 (Tamura et al., 2013). Mean uncorrected genetic distances (p-distances) between sequences were determined using MEGA 6.0. MODELTEST v.3.06 (Posada & Crandall, 1998) was applied to estimate the optimal evolutionary models to be used for the data set analysis. The best-fitting model was determined to be the (GTR + I + G) model of DNA evolution, as suggested by the Akaike information criterion. Both 12S rRNA and 16S rRNA gene fragments were treated as a single partition due to the relatively short sequence length and similar features (i.e., mitochondrial rRNA).

Phylogenetic trees were inferred using two different methods: maximum likelihood (ML) and Bayesian inference (BI). The ML analysis was conducted using Treefinder (Jobb, von Haeseler & Strimmer, 2004). Confidence in tree topology was evaluated by nonparametric bootstrap (BS) analysis with 1,000 replicates (Felsenstein, 1985). The BI analysis was conducted using MrBayes 3.1.2 (Huelsenbeck & Ronquist, 2001; Ronquist & Huelsenbeck, 2003); Metropolis-coupled Markov chain Monte Carlo (MCMCMC) analyses were run with one cold chain and three heated chains for four million generations and were sampled every 1,000 generations. Five independent MCMCMC runs were performed and 1,000 trees were discarded as burn-in. Confidence in the tree topology was assessed using posterior probability (PP) (Huelsenbeck & Ronquist, 2001). We a priori regarded tree nodes with BS values of 75% or greater and PP values over 0.95 as sufficiently resolved, those with BS values in the range between 75% and 50% (PP between 0.95 and 0.90) were regarded as tendencies, those with BS below 50% (PP below 0.90) were considered to be unresolved (Huelsenbeck & Hillis, 1993).

Adult morphology

Sex of adult individuals was determined using gonadal dissection. All measurements were taken to the nearest 0.02 mm (and subsequently rounded to a 0.1 mm precision) from preserved specimens using digital caliper under a light dissecting microscope; morphometrics were acquired according to Poyarkov et al. (2014): (1) snout–vent length (SVL; measured from the tip of the snout to cloaca); (2) head length (HL; measured from the tip of snout to hind border of jaw angle); (3) snout length (SL; measured from the anterior corner of eye to the tip of snout); (4) eye length (EL; measured as the distance between anterior and posterior corners of the eye); (5) nostril–EL (N–EL; measured as the distance between the anterior corner of the eye and the nostril center); (6) head width (HW; measured as the maximum width of head on the level of mouth angles in ventral view); (7) internarial distance (IND; measured as the distance between the central points of nostrils); (8) interorbital distance (IOD; measured as the shortest distance between the medial edges of eyeballs in dorsal view); (9) upper eyelid width (UEW; measured as the maximum distance between the medial edge of eyeball and the lateral edge of upper eyelid); (10) forelimb length (FLL; measured as the length of straightened forelimb to the tip of third finger); (11) lower arm and hand length (LAL; measured as the distance between elbow and the tip of third finger); (12) hand length (HAL; measured as the distance between the proximal end of outer palmar (metacarpal) tubercle and the tip of third finger); (13) inner palmar tubercle length (IPTL; measured as the maximum distance between proximal and distal ends of inner palmar tubercle); (14) outer palmar tubercle length (OPTL; measured as the maximum diameter of outer palmar tubercle); (15) hindlimb length (HLL; measured as the length of straightened hindlimb from groin to the tip of fourth toe); (16) tibia length (TL; measured as the distance between the knee and tibiotarsal articulation); (17) foot length (FL; measured as the distance between the distal end of tibia and the tip of fourth toe); (18) inner metatarsal tubercle length (IMTL; measured as the maximum length of inner metatarsal tubercle); (19) first toe length (1TOEL), measured as the distance between the distal end of inner metatarsal tubercle and the tip of first toe; (20–23) second to fifth toe lengths (measured as the outer lengths for toes II–IV, as the inner length for toe V; 2–5TOEL); (24) first finger width (1FW), measured at the distal phalanx; (25–27) finger disk diameters (2–4FDW); (28–32) toe disk diameters (1–5TDW); (33–36) finger lengths (1–3FLO, 4FLI; for outer side (O) of the first, inner side (I) of the fourth, measured as the distance between the tip and the junction of the neighboring finger); (37) Tympanum length, measured as the maximum tympanum diameter (TMP); (38) Tympanum-eye distance (TEY). Terminology for describing eye coloration in living individuals is in accordance with Glaw & Vences (1997); subarticular tubercle formulas follow those of Savage (1975).

The morphological characters for comparison and the data on their states in other Microhylidae representatives were taken from the following studies: Burton (1986), Chan et al. (2009), Günther & Richards (2016), Günther (2009, 2017), Günther, Stelbrink & von Rintelen (2010, 2012), Günther et al. (2012), Günther, Richards & Dahl (2014), Günther, Richards & Tjaturadi (2016), Köhler & Günther (2008), Kraus & Allison (2003), Kraus (2010, 2011, 2013a, 2013b, 2014, 2016, 2017), Menzies & Tyler (1977), Parker (1934), Richards & Iskandar (2000), Richards, Johnston & Burton (1992, 1994), Rittmeyer et al. (2012), Zweifel (1972, 2000), Zweifel, Menzies & Price (2003).

Larval morphology

Morphological description of larval stages follows Poyarkov et al. (2015, 2017), and Vassilieva et al. (2014, 2017) and includes the following 16 measurements: total length (ToL); body length (BL); tail length (TaL); maximum body width (BW); maximum body height (BH); maximum tail height (TH); SVL; snout to spiracle distance (SSp); maximum upper tail fin height (UF); maximum lower tail fin height (LF); internarial distance (IND); interpupilar distance (IP); rostro-narial distance (RN); naro-pupilar distance (NP); eye diameter (ED), and mouth width (MW). Tadpoles were staged according to the table of Gosner (1960).

Osteology

MicroCT scanning protocols followed Scherz et al. (2016). MicroCT scanning was conducted at the Petroleum Geology Department, Faculty of Geology, Lomonosov Moscow State University using a SkyScan 1172 desktop scanner (Bruker microCT, Kontich, Belgium) equipped with a Hamamatsu 10Mp digital camera. Specimen was mounted on a polystyrene baseplate and placed inside a hermetically closed polyethylene vessel. Scans were conducted with a resolution of 3.7 μm at 100 keV voltages and current of 100 mA with rotation step 0.2° with the use of oversize mode in which four blocks of subscan data were connected vertically to obtain a general tomogram. Data processing was performed using Skyscan software: NRecon (reconstruction) and CTan/CTVol (3D model producing and imaging). Osteological terminology follows Scherz et al. (2016) and Trueb (1968, 1973). MicroCT does not render cartilage, cartilage structures were therefore omitted from the osteological descriptions.

Results

Sequence variation

The studied 12S rRNA–16S rRNA mtDNA fragment consisted of 2,591 sites: 1,070 sites were conserved and 1,394 sites were variable, 1,070 of which were found to be parsimony-informative. Hypervariable regions with poor local alignment were removed using Gblocks v0.91b (Castresana, 2000); of the original 1,556 aligned positions, 2,253 were retained in final analyses. The transition–transversion bias (R) was estimated to be equal to 2.18. Nucleotide frequencies were A = 34.23%, T = 22.89%, C = 24.85%, and G = 18.04% (all data given only for the Microhylidae ingroup).

Phylogenetic relationships

Results of phylogenetic analyses are shown in Fig. 2. Bayesian and ML analyses yielded essentially similar topologies that slightly differed only in associations at several poorly supported basal nodes. We achieved high resolution of phylogenetic relationships among major lineages of the subfamily Asterophryinae, with several sufficiently resolved major nodes (PP = 1.0; BS = 100%; Fig. 2). However, phylogenetic relationships between the subfamilies of Microhylidae, or within the Austro–Papuan radiation of Asterophryinae were poorly resolved with low or insignificant levels of support (BPP < 0.95; BS < 75%) for major basal nodes.

Figure 2 Bayesian inference dendrogram of Asterophryinae derived from the analysis of 2,591 bp of the 12S rRNA–16S rRNA mtDNA gene fragment.

Voucher specimen IDs and GenBank accession numbers are given in Table 2. Sequence of Rhacophorus schlegelii is used as an outgroup. Numbers near the branches represent posterior probability (PP) or bootstrap support values (BS, 1,000 replicates) for the BI/ML inferences, respectively. “A” denotes the subfamily Asterophryinae sensu lato node. Thumbnails show adult specimens of Siamophryne troglodytes Gen. et sp. nov. and Gastrophrynoides sp. (Malaysia); photos by N. A. Poyarkov and Yu Lee.

The general topology of the phylogenetic relationships of microhylid frogs resulting from our analyses is consistent with the results reported in recent studies by Matsui et al. (2011), Kurabayashi et al. (2011), Pyron & Wiens (2011), de Sá et al. (2012), the mtDNA dataset of Peloso et al. (2016) and Rivera et al. (2017). Asterophryinae generic taxonomy is currently in a state of flux; we use the taxonomy proposed by Rivera et al. (2017), who synonymized genera Genyophryne, Metamagnusia, Pseudocallulops, Liophryne, and Oxydactyla, but also provide traditional generic affiliation for these groups in brackets (see Fig. 2).

The BI tree (Fig. 2) suggests the following set of genealogical relationships among the assessed microhylid taxa. Phylogenetic relationships among the subfamilies of Microhylidae are essentially unresolved; monophyly of the Dyscophinae, Kalophryninae, and Asterophryinae subfamilies is well-supported (1.0/100; hereafter, the node support values are given for BI PP/ML BS, respectively).

Asterophryinae consists of two major well-supported (1.0/100) reciprocally monophyletic clades:

(1) Asterophryinae 1, or “core” Asterophryinae, includes all presently known genera of the subfamily that inhabit Australasia east of the Wallace line and the island of Bali (see line B1 on Fig. 1; range of Asterophryinae 1 is marked in red). Phylogenetic relationships among genera within the clade Asterophryinae 1 remain essentially unresolved (Fig. 2); they are discussed in more details in a multilocus study of Rivera et al. (2017).

(2) The second clade includes the genus Gastrophrynoides known to date only from Sundaland—Peninsular Malaysia and Borneo (lineage Asterophryinae 2 on Fig. 2; range on Fig. 1 is marked in blue), and the newly discovered microhylid from Kanchanaburi Province in western Thailand (lineage Asterophryinae 3 on Fig. 2; locality on Fig. 1 is marked in green).

Thus, our phylogenetic analyses indicate that the newly discovered Microhylidae Gen. sp. from Kanchanaburi Province in western Thailand falls into the radiation of Asterophryinae sensu lato and is placed as a sister lineage to the genus Gastrophrynoides with high levels of node support.

Genetic distances

The uncorrected genetic p-distances between the 12S rRNA–16S rRNA gene fragments among and within the studied Microhylidae genera are shown in the Table 3. The genetic differentiation between the Microhylidae Gen. sp. from Kanchanaburi Province and other Microhylidae genera vary from 14.8% (genus Liophryne) to 20.6% of substitutions (genus Barygenys). Genetic distance between the Microhylidae Gen. sp. and its sister lineage Gastrophrynoides reaches 15.6% of substitutions. These values of genetic divergence are high and correspond well to the genus level of differentiation observed in other groups of Anura (Vences et al., 2005a, 2005b; Vieites et al., 2009). No genetic variation was recorded in obtained haplotypes of 12S rRNA–16S rRNA gene of the new species (Table 3).

Table 3 Genetic divergence of Siamophryne troglodytes Gen. et sp. nov Uncorrected p-distances (percentage) between 12S rRNA and 16S rRNA sequences of Siamophryne troglodytes Gen. et sp. nov. and other Microhylidae genera included in phylogenetic analyses (below the diagonal line), and standard error estimates (above the diagonal line).

	Genus	1	2	3	4	5	6	7	8	9	10	11	12	13	14	15	16	17	18	19	20	21	22	
1	Siamophryne troglodytes Gen. et sp. nov.	0.0	1.4	1.6	1.6	1.5	1.7	1.5	1.6	1.6	1.5	1.5	1.4	1.4	1.5	1.5	1.5	1.3	1.5	1.7	1.6	1.5	2.0	
2	Gastrophrynoides	15.6	–	1.5	1.4	1.4	1.5	1.5	1.4	1.5	1.6	1.3	1.4	1.3	1.5	1.4	1.5	1.2	1.4	1.4	1.5	1.5	1.7	
3	Aphantophryne	15.9	16.0	–	1.3	1.1	1.5	1.3	1.4	1.3	1.4	1.3	1.4	1.2	1.4	1.3	1.4	0.9	1.4	1.5	1.4	1.4	1.5	
4	Asterophrys	17.5	18.0	15.0	–	1.2	1.5	1.4	1.4	1.4	1.5	1.5	1.5	1.4	1.5	1.1	1.3	1.1	1.4	1.6	1.2	1.4	1.4	
5	Austrochaperina	16.3	17.3	12.7	14.4	12.7	1.2	1.2	1.3	1.3	1.2	1.3	1.2	1.1	1.1	1.2	1.3	0.9	1.2	1.3	1.3	1.1	1.3	
6	Barygenys	20.6	18.1	15.6	15.7	14.7	–	1.7	1.6	1.5	1.4	1.5	1.5	1.4	1.5	1.5	1.4	1.2	1.5	1.6	1.7	1.4	1.6	
7	Callulops	17.4	16.1	13.5	13.4	14.0	16.8	–	1.4	1.3	1.5	1.5	1.3	1.3	1.2	1.2	1.4	1.2	1.3	1.5	1.4	1.4	1.5	
8	Choerophryne	19.4	19.2	15.7	17.9	16.9	18.9	16.1	–	1.3	1.6	1.3	1.5	1.4	1.6	1.3	1.4	1.2	1.5	1.5	1.4	1.4	1.7	
9	Cophixalus	17.0	17.6	13.4	15.7	15.2	15.1	14.7	15.9	–	1.5	1.3	1.4	1.4	1.3	1.2	1.4	1.1	1.1	1.5	1.4	1.2	1.5	
10	Copiula	16.8	19.6	15.5	16.0	14.0	18.9	17.5	19.1	17.5	–	1.5	1.5	1.5	1.4	1.4	1.5	1.2	1.6	1.6	1.4	1.5	1.8	
11	Genyophryne	16.0	17.2	12.5	16.7	14.3	15.2	15.9	15.2	14.1	17.2	–	1.4	1.4	1.3	1.5	1.5	1.2	1.6	1.5	1.5	1.3	1.8	
12	Hylophorbus	16.5	18.2	15.7	16.4	16.0	17.9	12.9	16.2	16.4	16.3	16.4	–	1.3	1.3	1.3	1.4	1.1	1.3	1.5	1.4	1.6	1.9	
13	Liophryne	14.8	15.9	11.7	13.0	12.4	14.5	11.4	16.4	13.2	15.5	14.2	14.0	–	1.2	1.2	1.4	1.0	1.3	1.4	1.4	1.2	1.5	
14	Mantophryne	16.5	17.4	15.7	14.4	13.9	17.1	12.7	16.8	14.6	16.2	14.9	12.0	12.9	–	1.2	1.2	1.1	1.3	1.3	1.3	1.4	1.7	
15	Metamagnusia	16.6	16.5	14.5	7.3	13.8	16.6	12.9	16.6	14.0	15.0	15.2	15.4	12.7	12.2	–	1.2	1.0	1.3	1.4	1.2	1.2	1.5	
16	Oninia	16.7	17.7	14.7	16.7	16.6	17.9	17.3	16.9	17.4	18.1	16.4	16.7	15.1	16.2	14.7	–	1.1	1.5	1.7	1.3	1.3	1.6	
17	Oreophryne	17.8	17.7	14.2	15.7	14.5	16.2	15.5	17.7	16.1	16.9	16.4	16.2	13.5	16.4	14.7	17.2	15.1	1.0	1.1	1.1	1.1	1.2	
18	Oxydactyla	16.1	15.7	14.4	13.7	12.2	15.4	12.2	15.9	11.2	16.5	14.7	14.7	9.2	13.7	12.7	15.7	14.4	–	1.4	1.5	1.1	1.7	
19	Paedophryne	17.9	17.7	16.2	16.7	15.4	16.9	17.9	20.1	17.1	18.4	17.3	19.4	16.6	17.4	14.7	18.8	16.5	15.9	–	1.4	1.4	1.6	
20	Pseudocallulops	17.5	17.7	15.7	12.9	15.6	18.2	14.5	16.6	16.6	14.8	16.2	15.5	15.5	15.1	12.7	16.6	16.8	16.0	16.2	–	1.4	1.5	
21	Sphenophryne	17.2	19.4	14.2	15.2	14.2	15.4	13.9	17.6	15.1	16.5	14.7	16.7	10.2	13.5	13.2	16.4	15.1	11.2	16.2	17.2	–	1.4	
22	Xenorhina	19.5	20.4	16.4	14.9	15.6	18.9	15.6	19.5	19.0	17.4	19.4	18.1	15.9	16.8	14.1	19.0	18.3	16.9	18.7	15.5	16.7	14.0	
Note:

The mean uncorrected p-distances within those genera for which more than one specimen was examined are shown in the diagonal and shaded with gray.

Taxonomy

Based upon the results of phylogenetic analyses of 12S rRNA–16S rRNA mtDNA fragment sequences, the Microhylidae frog from Kanchanaburi Province represents a previously unknown highly divergent mtDNA lineage, clearly distinct from all other members of Microhylidae for which comparable genetic data were available. This lineage falls into the Australasian subfamily Asterophryinae and with high values of node support is reconstructed as a sister group to the genus Gastrophrynoides that inhabits Borneo and the Peninsular Malaysia. Subsequent analyses of osteology and external morphology (see below) clearly indicate that the recently discovered population of Microhylidae Gen. sp. from Kanchanaburi Province represents a new previously undescribed genus and species which we describe herein as: Amphibia Linnaeus, 1758

Anura Fischer von Waldheim, 1813

Microhylidae Günther, 1858

Asterophryinae Günther, 1858

Siamophryne Gen. nov.

Diagnosis

A medium-sized (19 mm < SVL < 30 mm) member of the Australasian subfamily Asterophryinae (family Microhylidae), with the following combination of morphological attributes: (1) both maxillae and dentaries eleutherognathine, no maxillary teeth; (2) vertebral column procoelous with eight presacral vertebrae (PSV) lacking neural crests; (3) no sagittal crest on cranium; (4) frontoparietals conjoined, connected by long suture; (5) nasals wide, calcified, but not contacting each other medially; (6) vomeropalatines small, not expanded, vomerine spikes absent; (7) cultriform process of parasphenoid comparatively narrow; (8) clavicles present as slender tiny bones, lying on the procoracoid cartilage not reaching scapula or the midline; (9) omosternum absent; (10) sternum large, anterior portion consists of calcified cartilage, xiphisternum cartilaginous; (11) weak dorsal crest present on urostyle, absent on ilium; (12) terminal phalanges large T-shaped; (13) all fingers and toe discs with terminal grooves; (14) subarticular tubercles weak, discernible only at digit basis; (15) toe webbing absent; (16) tympanum distinct; (17) two transverse smooth palatal folds; (18) pupil round; (19) snout rounded, equal to EL; (20) development with a larval stage, tadpole with peculiar dorso-ventrally compressed morphology.

Type species. Siamophryne troglodytes sp. nov.

Other included species. None are known at present.

Distribution

To date, S. troglodytes sp. nov. is only known from a small cave system in a karst region of Sai Yok District, Kanchanaburi Province, northern Tenasserim Region, western Thailand (see below the description of the species) (see Fig. 1).

Discrimination from other Asterophryinae genera

Information on character states for other Asterophryinae genera is based on Parker (1934), Zweifel (1972, 2000), Menzies & Tyler (1977), Burton (1986), Zweifel, Menzies & Price (2003), Günther, Stelbrink & von Rintelen (2010), Kraus (2010, 2017), and references therein. The new genus has eleutherognathine maxillae and dentaries and thereby is distinguished from those Asterophryinae genera which have symphignathine state of this trait in both jaws: Asterophrys (including the recently synonymized Pseudocallulops and Metamagnusia; see Rivera et al., 2017) (New Guinea), Callulops (from Sulawesi to New Guinea region), Mantophryne (New Guinea and Louisiade Archipelago), Oninia (New Guinea), and Xenorhina (including the recently synonymized Xenobatrachus Peters & Doria) (New Guinea region). The genus Barygenys (Papua New Guinea region) can be distinguished from the new genus by the presence of symphignathine dentaries and eleutherognathine maxillae (vs. both jaws being eleutherognathine in Siamophryne Gen. nov.). The new genus lacks distinct neural crests on PSV, and therefore can be differentiated from the genera Aphantophryne (from Philippines to New Guinea) and Cophixalus (from Moluccas to New Guinea and northern Australia) (both have well-developed neural crests on PSV); the new genus can be further distinguished from Aphantophryne as it has eight PSV (vs. seven PSV in Aphantophryne). The genus Sphenophryne sensu lato (New Guinea) has well-developed long and slender clavicles (vs. tiny clavicles that do not reach scapula and the midline in the new genus), and broad vomeropalatines that contact each other medially, with a postchoanal portion overlying the palatine region (vs. vomeropalatines not expanded in the new genus); Sphenophryne sensu stricto (S. cornuta Peters & Doria) can be further distinguished by a characteristic spine-like projection on the upper eyelid (vs. smooth upper eyelid in the new genus), it also has arboreal life style (vs. troglophilous life style of the new genus). The genus Genyophryne (recently considered as a synonym of Sphenophryne; see Rivera et al., 2017) can be distinguished from the new genus by absence of clavicles (vs. clavicles present), stout body habitus (vs. slender body habitus) and absence of large finger discs (vs. very large finger discs in the new genus). The genus Liophryne (which is also regarded as a member of Sphenophryne sensu lato based on phylogenetic data of Rivera et al., 2017) can be differentiated from the new genus by the presence of long and slender clavicles (vs. tiny clavicles in the new genus), and by the presence of comparatively small finger discs (vs. large broad fingers discs in the new genus). The genus Oxydactyla (now regarded as a part of Sphenophryne sensu lato; see Rivera et al., 2017) can be distinguished from the new genus by the absence of finger discs (vs. large finger discs present in the new genus). The genus Paedophryne (New Guinea region) can be distinguished from the new genus by a much smaller body size (SVL < 20 mm vs. SVL ≥ 20 mm in the new genus), by cartilaginous phalanges in the first digit (vs. ossified phalanges in the new genus), by FI reduced to a nub (vs. well-developed FI in the new genus), by the absence of clavicles and procoracoids (vs. presence in the new genus) and by having seven PSV (vs. eight PSV in the new genus). By the presence of clavicles and procoracoid cartilage, the new genus can be differentiated from the genus Choerophryne (New Guinea), which lacks these structures; the latter also has palatine portions of vomeropalatines fused with broad sphenethmoids (vs. not fused in the new genus). The genus Copiula (New Guinea) can be distinguished from the new genus by the lack of clavicles (vs. presence in the new genus), by small discs on fingers, which are smaller than those on toes (vs. large finger discs that are larger than toe discs in the new genus), by cartilaginous sternum (vs. ossified anterior portion of sternum in the new genus), and by the presence of a conspicuous rostral dermal gland (vs. rostral gland absent in the new genus). The new genus can be distinguished from Austrochaperina (Australia, New Guinea, and New Britain) by fingers with very wide discs, much wider than penultimate phalanges, by vomeropalatines not expanded and by narrow cultriform process of parasphenoid (vs. discs on fingers absent or small, slightly different in width from penultimate phalanges, vomeropalatines expanded and broad cultriform process of parasphenoid in Austrochaperina). The new genus can be distinguished from Hylophorbus (New Guinea) by comparatively better developed nasals (vs. poorly developed nasals in Hylophorbus), by the presence of large finger discs (vs. discs on fingers usually absent, if present, they are much smaller than toe discs) and by completely smooth skin on dorsum (vs. shagreened to tubercular skin on dorsum in Hylophorbus). The genus Oreophryne (from Philippines and Lesser Sundas to New Guinea and New Britain, see Kraus, 2017) can be differentiated from the new genus by distinct toe webbing (vs. no toe webbing in the new genus), by arboreal or terrestrial life style (vs. troglophilous life style in the new genus), and by expanded vomeropalatines (vs. not expanded in the new genus).

From its sister genus Gastrophrynoides (Peninsular Malaysia and Borneo) the new genus can be easily distinguished by the presence of large and wide finger discs (vs. small finger discs, slightly wider than the penultimate phalanges in Gastrophrynoides), by a comparatively much shorter snout and larger eye (SL equal to EL in the new genus vs. snout 2.5 times longer than eye in Gastrophrynoides) and by a distinct tympanum (vs. tympanum obscured by skin in Gastrophrynoides).

Finally, the sequences of the 12S rRNA–16S rRNA mtDNA fragment for the new genus are markedly distinct from the sequences for all those Microhylidae members, for which homologous sequences are available (see Fig. 2; Table 3).

Etymology

The generic nomen Siamophryne is derived from “Siam”—the old name of present-day Thailand; referring to the range of the new genus, which to date is only known from western Thailand; and the Greek noun “phryne” (φρÚνη; feminine gender), meaning “toad” in English; this root is often used in the generic names in Asterophryinae microhylid frogs. Gender of the new genus is feminine.

Siamophryne troglodytes sp. nov.

Figs. 3–10; Table 4.

Figure 3 The male holotype of Siamophryne troglodytes Gen. et sp. nov. (AUP-00500) in preservative

(A) Dorsal view; (B) ventral view; (C) lateral view of the head; (D) volar view of the right hand; (E) plantar view of the right foot. Photos by N. A. Poyarkov.

Table 4 Measurement data for Siamophryne troglodytes Gen. et sp. nov. type series.

Characters	Holotype	Paratype males	Paratype females	
Museum ID	AUP-00500	AUP-00501	AUP-00502	AUP-00503	AUP-00504	NAP-06651	Males total (n = 6)	AUP-00505	AUP-00506	AUP-00507	AUP-00508	NAP-06652	Females total (n = 5)	
Sex	m	m	m	m	m	m	Mean ± SD	Min–Max	f	f	f	f	f	Mean ± SD	Min–Max	
(1) SVL	22.1	24.9	19.8	24.6	24.1	19.1	22.4 ± 2.1	(19.1–24.9)	26.6	25.9	25.0	27.8	26.1	26.3 ± 0.7	(25.0–27.8)	
(2) HL	6.7	7.7	6.3	7.4	7.4	6.3	7.0 ± 0.5	(6.3–7.7)	8.1	7.8	7.6	8.8	8.1	8.1 ± 0.3	(7.6–8.8)	
(3) SL	2.7	2.7	2.4	2.9	2.8	2.2	2.6 ± 0.2	(2.2–2.9)	3.0	2.9	3.0	3.3	3.0	3.0 ± 0.1	(2.9–3.3)	
(4) EL	3.0	3.2	2.5	3.0	2.8	2.4	2.8 ± 0.2	(2.4–3.2)	3.3	3.6	3.2	3.4	3.0	3.3 ± 0.2	(3.0–3.6)	
(5) N–EL	1.9	1.5	1.5	1.9	2.0	1.2	1.7 ± 0.3	(1.2–2.0)	2.0	1.6	2.1	2.2	1.7	1.9 ± 0.2	(1.6–2.2)	
(6) HW	7.1	7.6	6.6	8.6	7.5	6.6	7.3 ± 0.6	(6.6–8.6)	8.5	7.7	9.4	8.1	8.8	8.5 ± 0.5	(7.7–9.4)	
(7) IND	2.3	2.3	1.9	2.4	2.4	1.7	2.2 ± 0.2	(1.7–2.4)	2.6	2.7	2.5	2.8	2.7	2.6 ± 0.1	(2.5–2.8)	
(8) IOD	2.2	2.4	1.4	2.4	2.5	2.0	2.1 ± 0.3	(1.4–2.5)	2.4	2.3	2.5	2.8	2.5	2.5 ± 0.1	(2.3–2.8)	
(9) UEW	1.8	2.1	1.5	1.8	1.8	1.2	1.7 ± 0.2	(1.2–2.1)	2.1	2.1	2.0	2.1	2.0	2.0 ± 0.1	(2.0–2.1)	
(10) FLL	15.4	16.9	13.9	18.3	16.2	13.5	15.7 ± 1.3	(13.5–18.3)	17.9	18.2	17.9	18.1	17.5	17.9 ± 0.2	(17.5–18.2)	
(11) LAL	10.4	11.6	9.6	11.9	11.3	9.9	10.8 ± 0.8	(9.6–11.9)	14.2	12.0	12.2	12.3	13.0	12.7 ± 0.7	(12.0–14.2)	
(12) HAL	5.6	6.7	4.9	6.4	6.0	5.5	5.9 ± 0.5	(4.9–6.7)	6.7	7.0	6.8	6.8	7.4	6.9 ± 0.2	(6.7–7.4)	
(13) IPTL	1.0	1.1	0.9	1.0	1.0	0.5	0.9 ± 0.1	(0.5–1.1)	1.0	0.8	1.0	1.0	1.0	1.0 ± 0.1	(0.8–1.0)	
(14) OPTL	1.1	1.0	1.0	1.0	1.1	0.7	1.0 ± 0.1	(0.7–1.1)	1.0	1.1	1.1	1.1	0.6	1.0 ± 0.1	(0.6–1.1)	
(15) HLL	37.1	35.4	30.3	36.9	34.4	29.9	34.0 ± 2.3	(29.9–37.1)	41.7	37.0	37.1	38.6	38.1	38.5 ± 1.3	(37.0–41.7)	
(16) TL	10.8	10.1	10.0	12.4	11.5	9.7	10.7 ± 0.8	(9.7–12.4)	13.3	10.2	12.8	13.2	12.3	12.4 ± 0.9	(10.2–13.3)	
(17) FL	9.6	10.9	7.9	10.4	10.7	9.5	9.8 ± 0.8	(7.9–10.9)	11.9	11.8	12.5	11.7	13.2	12.2 ± 0.5	(11.7–13.2)	
(18) IMTL	0.9	0.9	0.8	1.0	1.0	0.4	0.8 ± 0.1	(0.4–1.0)	1.5	1.5	1.5	1.5	1.0	1.4 ± 0.2	(1.0–1.5)	
(19) 1TOEL	2.0	2.1	2.9	3.2	2.4	1.3	2.3 ± 0.5	(1.3–3.2)	2.9	2.2	2.5	2.7	1.8	2.4 ± 0.3	(1.8–2.9)	
(20) 2TOEL	4.7	4.6	4.7	5.7	4.9	2.4	4.5 ± 0.7	(2.4–5.7)	5.8	4.6	4.5	4.5	4.0	4.7 ± 0.4	(4.0–5.8)	
(21) 3TOEL	7.4	7.5	7.1	8.1	8.2	3.9	7.0 ± 1.0	(3.9–8.2)	9.3	8.6	8.2	7.6	5.6	7.8 ± 1.0	(5.6–9.3)	
(22) 4TOEL	9.6	10.9	7.9	10.4	10.7	8.7	9.7 ± 1.0	(7.9–10.9)	11.9	11.8	12.5	11.7	12.5	12.1 ± 0.3	(11.7–12.5)	
(23) 5TOEL	7.3	6.6	6.3	7.6	7.8	2.8	6.4 ± 1.2	(2.8–7.8)	8.7	7.6	7.7	7.1	8.8	8.0 ± 0.6	(7.1–8.8)	
(24) 1FW	1.0	1.0	0.7	1.0	0.9	0.5	0.8 ± 0.2	(0.5–1.0)	0.8	1.0	1.1	1.1	0.6	0.9 ± 0.2	(0.6–1.1)	
(25) 2FDW	1.2	1.3	1.0	1.3	1.0	0.7	1.1 ± 0.1	(0.7–1.3)	1.6	1.6	1.3	1.4	1.2	1.4 ± 0.1	(1.2–1.6)	
(26) 3FDW	1.3	1.3	1.2	1.2	1.1	1.0	1.2 ± 0.1	(1.0–1.3)	1.5	1.5	1.4	1.6	1.3	1.4 ± 0.1	(1.3–1.6)	
(27) 4FDW	1.3	1.4	1.2	1.3	1.2	0.8	1.2 ± 0.1	(0.8–1.4)	1.3	1.6	1.4	1.5	1.1	1.4 ± 0.1	(1.1–1.6)	
(28) 1TDW	0.2	0.2	0.2	0.2	0.2	0.3	0.2 ± 0.0	(0.2–0.3)	0.2	0.2	0.2	0.2	0.5	0.3 ± 0.1	(0.2–0.5)	
(29) 2TDW	0.2	0.2	0.2	0.2	0.2	0.8	0.3 ± 0.1	(0.2–0.8)	0.2	0.2	0.2	0.2	0.8	0.3 ± 0.1	(0.2–0.8)	
(30) 3TDW	0.3	0.3	0.3	0.2	0.2	0.7	0.3 ± 0.1	(0.2–0.7)	0.2	0.2	0.3	0.2	1.0	0.4 ± 0.1	(0.2–1.0)	
(31) 4TDW	0.5	0.6	0.5	0.6	0.6	0.6	0.6 ± 0.0	(0.5–0.6)	0.6	0.6	0.6	0.7	1.1	0.7 ± 0.2	(0.6–1.1)	
(32) 5TDW	0.3	0.3	0.3	0.3	0.3	0.5	0.3 ± 0.1	(0.3–0.5)	0.3	0.3	0.3	0.4	0.6	0.4 ± 0.1	(0.3–0.6)	
(33) 1FLO	1.9	2.2	2.1	3.3	3.2	1.3	2.3 ± 0.5	(1.3–3.3)	2.9	2.8	1.9	2.8	1.9	2.4 ± 0.4	(1.9–2.9)	
(34) 2FLO	3.7	4.8	3.5	5.7	5.2	2.4	4.2 ± 0.9	(2.4–5.7)	4.7	4.7	3.8	4.0	3.8	4.2 ± 0.4	(3.8–4.7)	
(35) 3FLO	5.6	6.4	5.0	6.9	6.7	3.2	5.6 ± 1.0	(3.2–6.9)	6.6	6.7	5.2	6.2	4.3	5.8 ± 0.8	(4.3–6.7)	
(36) 4FLI	4.5	4.9	3.9	6.0	6.2	2.6	4.7 ± 1.0	(2.6–6.2)	5.5	4.8	3.2	5.4	3.1	4.4 ± 1.0	(3.1–5.5)	
(37) TMP	1.2	1.5	1.1	1.4	1.3	1.1	1.3 ± 0.1	(1.1–1.5)	1.4	1.5	1.6	1.1	1.6	1.4 ± 0.2	(1.1–1.6)	
(38) TEY	0.8	0.7	0.6	0.6	0.7	0.6	0.7 ± 0.1	(0.6–0.8)	0.6	0.7	1.0	0.6	0.6	0.7 ± 0.1	(0.6–1.0)	
Notes:

For other abbreviations see “Materials and Methods.” All measurements are in mm.

m, male; f, female; SD, standard deviation; n, number of measured specimens.

Holotype

AUP-00500, adult male in a good state of preservation, collected in a limestone cave in Sai Yok District, Kanchanaburi Province, western Thailand, elevation 440 m a.s.l. (approximately in the vicinity of 14°28′N, 98°51′E; exact geographic coordinates not provided for conservation purposes) (see Fig. 10); collected on October 27, 2016, by Montri Sumontha, Jitthep Tunprasert, Nirut Chomngam, and Chatmongkon Suwannapoom.

Paratypes

In total, 10 specimens: AUP-00501-00504, four adult males, and AUP-00505-00508, three adult females, collected on October 27, 2016, from the same locality and with the same data as the holotype; ZMMU A-5818 (field ID NAP-06651), adult male, and ZMMU A-5819 (field ID NAP-06652), adult female, collected by T. Ruangsuwan from the same locality as the holotype on August 1, 2016.

Referred specimens

AUP-00509, a larval specimen, Gosner stage 36, collected by T. Ruangsuwan from the same locality as the holotype on August 1, 2016.

Diagnosis

The only known member of the genus Siamophryne Gen. nov. (see Diagnosis of the genus). S. troglodytes sp. nov. is characterized by a combination of the following traits: (1) SVL of six adult males 19.1–24.9 mm, and of five adult females 25.0–27.8 mm; (2) body habitus slender, limbs very long (FLL/SVL ratio 0.69 (0.65–0.74); HLL/SVL 1.50 ratio (1.39–1.67) for both sexes); (3) snout short, rounded, subequal to EL (0.8–1.0 times the length of the eye); (4) eye medium-sized, EL/SVL ratio (0.12–0.14); (5) tips of fingers II–IV expand to broad discs 1.5–2.5 times wider than the penultimate phalanges; toes II–IV with smaller discs slightly wider than the penultimate phalanges, tips of finger I, toe I, and toe V rounded, same width as the penultimate phalanges; (6) finger discs distinctly wider than toe discs; (7) terminal phalanges distinctly T-shaped in F2–F4 and T2–T5; bobbin-shaped in finger I and toe I; (8) subarticular tubercles on fingers weak, indistinct; finger subarticular tubercle formula: 1:1:1:1; better pronounced on toes, toe subarticular tubercle formula: 1:1:1:1:1; (9) outer metatarsal tubercle absent, inner metatarsal tubercle small, rounded; (10) skin of the ventral surface completely smooth, skin of the dorsal and lateral surfaces smooth with rare flat tubercles or pustules; (11) osteological features are the same as for the genus. Other diagnostic features are given in the diagnosis of the genus.

Description of the holotype

Holotype in preservative is shown in Figs. 3 and 4. Medium-sized specimen, with SVL 22.1 (hereafter all measurements in mm), in a good state of preservation, however distal parts of toes II—V slightly dehydrated (Fig. 3E); ventral surface of left thigh dissected for 5 mm and some of femoral muscles removed. Body habitus slender (Fig. 3A); HL slightly shorter than HW (0.95); snout rounded both in profile (Figs. 3C and 4A) and in dorsal view (Fig. 3A), shorter than the diameter of eye (SL/EL 0.91); eyes large, notably protuberant in dorsal and lateral views, pupil oval, horizontal (Fig. 3C); dorsal surface of head flat, canthus rostralis indistinct, gently rounded; loreal region weakly concave; nostril rounded, lateral, located much closer to tip of snout than to eye; tympanum well discernable, circular, tympanic rim not elevated above the skin of temporal area, supratympanic fold absent; vomerine teeth absent, two transverse palatal folds present across the palate anteriorly to the pharynx, both of them with smooth edges, tongue spatulate and free behind, lacking papillae, vocal sac opening not discernable.

Figure 4 Morphological details of the male holotype of Siamophryne troglodytes Gen. et sp. nov. (AUP-00500) in preservative.

(A) Head in the lateral view; (B) volar view of the right hand; (C) plantar view of the right foot. Scale bar equals 2 mm. Drawings by Valentina D. Kretova.

Forelimbs comparatively long, less than half length of hindlimbs (FLL/HLL 0.42); hand slightly longer than lower arm and less than half length of forelimb (HAL/FLL 0.36); fingers slender, flattened in cross section, first finger well developed, one-half length of the second finger (1FLO/2FLO 0.50); relative finger lengths: I < II < IV < III (see Figs. 3D and 4B). Finger webbing and dermal fringes absent. First fingertip rounded and slightly dilated, almost the same width as the basal phalanx. Tips of three outer fingers II–IV greatly dilated forming large triangular disks with distinct narrow terminal grooves; relative finger disk widths: II < IV < III; longitudinal furrow on the dorsal surface of fingers absent; flexor tendons visible through translucent skin on ventral surface of fingers; subarticular tubercles on fingers barely distinct at basis of proximal phalanges and almost indistinct under penultimate phalanges of fingers III and IV, subarticular tubercles flat, oval-shaped with unclear borders, finger subarticular tubercle formula: 1:1:1:1 (for fingers I:II:III:IV, respectively); nuptial pad absent; two palmar (metacarpal) tubercles: inner palmar tubercle small, rounded, same size as subarticular tubercles; outer palmar tubercle oval-shaped with indistinct borders, almost the same length as inner palmar tubercle (IPTL/OPTL 0.97); palmar surface smooth, supernumerary palmar tubercles absent.

Hindlimbs long and slender, TL is half of SVL (0.49); tibiotarsal articulation of adpressed limb reaching the eye level; foot shorter than tibia (FL/TL 0.89); relative toe lengths I < II < V < III < IV; tarsus smooth, tarsal fold absent; tips of all toes slightly dilated forming small spatulate disks on all toes except toe I (Figs. 3E and 4C), each disk with weak terminal groove similar to that on fingers, relative toe disk widths: I < II < V < III < IV; toes slightly flattened in cross section, dermal fringes absent; toe webbing absent between all toes; subarticular tubercles on toes more distinct than on fingers, oval-shaped, elevated, toe subarticular tubercle formula: 1:1:1:1:1 (for toes I:II:III:IV:V, respectively); single metatarsal tubercle: inner metatarsal tubercle rounded, flattened.

Skin on dorsal and dorsolateral surfaces smooth with rarely scattered flat tubercles (Fig. 3A), tubercles getting larger and more prominent on dorsal surfaces of hindlimbs; dorsal surface of forelimbs smooth with few small tubercles on forearm; upper eyelids smooth; ventral sides of trunk, head and limbs completely smooth (Fig. 3B); dermal ridges or skin macroglands absent.

Measurements of holotype (in mm)

SVL 22.1; HL 6.7; SL 2.7; EL 3.0; N–EL 1.9; HW 7.1; IND 2.3; IOD 2.2; UEW 1.8; FLL 15.4; LAL 10.4; HAL 5.6; IPTL 1.0; OPTL 1.1; HLL 37.1; TL 10.8; FL 9.6; IMTL 0.9; 1TOEL 2.0; 2TOEL 4.7; 3TOEL 7.4; 4TOEL 9.6; 5TOEL 7.3; 1FW 1.0; 2FDW 1.2; 3FDW 1.3; 4FDW 1.3; 1TDW 0.2; 2TDW 0.2; 3TDW 0.3; 4TDW 0.5; 5TDW 0.3; 1FLO 1.9; 2FLO 3.7; 3FLO 5.6; 4FLI 4.5; TMP 1.2; TEY 0.8.

Coloration of holotype in life

Coloration rather uniform: dorsal surface of head and trunk chocolate brown; dorsal surface of limbs ochre-brown; ventral surface of limbs and throat light orange-pink, lateral sides of body gray to gray-brown; ventral surfaces of body pinkish-gray. Fingers and toes gray with dark brown mottling. Tympanum grayish with slight brown mottling. Tubercles on dorsal surfaces of body, limbs and head copper-brown. Iris uniform dark brown; pupil black; sclera bluish-gray.

Coloration of holotype in preservative

Coloration in preservative is shown on Fig. 3. After preservation in ethanol, dorsal coloration changed to grayish-brown, upper eyelids dark-gray (Fig. 3A), ventral surface of chest, belly, limbs turn yellowish-gray (Fig. 3B); iris coloration faded and turned black.

Morphological variation of the type series

Measurements of the type series that show variation in morphometric characteristics are given in Table 4. Coloration of paratype in life is shown in Fig. 5. Specimens show no significant variation in coloration. Specimens vary in the body size: females (SVL 25.0–27.8; mean 26.3 ± 0.7; n = 5) are larger than males (SVL 19.1–24.9; mean 22.4 ± 2.1; n = 6). Paratypes in situ had generally darker coloration with dark gray-brown coloration of dorsum and gray coloration of ventral surfaces and body flanks (Figs. 10C and 10D).

Figure 5 Male paratype of Siamophryne troglodytes Gen. et sp. nov. (ZMMU A-5818) in life in dorsolateral aspect.

Photo by N. A. Poyarkov.

Osteological characteristics

Based on microtomographic data from ZMMU A-5818, adult male. The main skeletal features are shown in Fig. 6. Details of skull morphology are presented in Fig. 7.

Figure 6 Osteology of Siamophryne troglodytes Gen. et sp. nov. (paratype, ZMMU A-5818).

The full skeleton is shown in (A) dorsal, and (B) ventral views; the right forelimb in (C) dorsal, and (D) volar aspects; and the right foot in (E) plantar, and (F) dorsal aspects. Digits numbered I–V. antbr., os antebrachii (radius + ulna); carp.d.II–IV, carpale distale F2–F4; centr., centrale; clav., clavicle; cor., coracoid bone; crur., os cruris (tibia + fibula); fem., femoral bone; fib., fibulare; hm., humeral bone; il., ilium; mtc.I–IV, metacarpalia F1–F4; mtt.I–V, metatarsalia T1–T5; ph.d.I–IV, finger phalanges F1–F4; ph.d.I–V, toe phalanges T1–T5; pr.p.-m., processus postero-medialis; prhl., prehallux; prpl., prepollex; prsac.v., presacral vertebrae; rad., radiale; sac.v., sacral vertebra; sc., scapula; strn., sternum; tar.d.II–III, tarsale distale T2–T3; tib., tibiale; uln., ulnare; ur., urostyle.

Figure 7 Head skeleton of Siamophryne troglodytes Gen. et sp. nov. (male paratype, ZMMU A-5818).

The skull is shown in (A) dorsal, (B) ventral, (C) lateral, and (D) frontal views. Scale bar equals to 1 mm. angspl., angulosplenial; col., columella; cond.oc., occipital condylus; dent., dentary bone; exoc., exoccipital; fpar., frontoparietal bone; max., maxilla; mmk., mentomeckelian bone; nas., nasal bone; pmax., premaxilla; proot., prootic; psph., parasphenoid; pter., pterygoid; qj., quadratojugal; smax., septomaxilla; spheth., sphenethmoid; sq., squamosal; vom., vomer.

Skull clearly longer than wide (Fig. 6). Frontoparietals separate along their entire length, longer than broad, narrower anteriorly than posteriorly, connected with each other medially with long suture, lack a sagittal crest, partially fused to exoccipital posteriorly (Fig. 7A). Exoccipitals separate, contact each other medially. Nasals large but not meeting each other at midline, lacking posterior ramus, chondrified peripherally, overlying the dorsal portion of sphenethmoid (Fig. 7A). Sphenethmoid ossified laterally and dorsally, but chondrified anteriorly and ventrally (Fig. 7B). Prootics partially chondrified, lacking dorsal crest (Fig. 7C). Squamosal L-shaped, well-ossified, distally chondrified, articulating on anterolateral surface of prootic (Fig. 7C). Columella comparatively small, centrally ossified (Fig. 7C), distally chondrified; tympanic annulus completely chondrified. Premaxilla with slender, weakly mineralized dorsal process; labial process of premaxilla well-ossified (Fig. 7D). Maxilla largely chondrified, ossified in central part. Anterior ends of maxillaries contact labial portions of well-developed premaxillaries (eleutherognathine condition) (Fig. 7D). Quadratojugal ossified in posterior portion. Vomers mostly chondrified plates meeting at midline, without teeth or lateral processes; parts edging choanae weakly ossified (Fig. 7C). Mentomeckelians ossified, connected to dentaries and to each other by strips of cartilage (Fig. 7B); dentaries not fused (eleutherognathine condition). Parasphenoid smooth; cultriform process of parasphenoid rather narrow, terminates at the level of sphenethmoid with an anterior notch (Fig. 7B). Hyoid plate completely cartilaginous, anterolateral (alary) processes of hyoid plate present, recurved, posterolateral processes thinner than alary processes; posteromedial processes strongly ossified, elongated, straight, wider at proximal ends, chondrified at distal ends (Fig. 6B).

Eight nonimbricate procoelous PSV, stout, length approximately from one-third to one-half of width; PSV longer anteriorly, narrowing progressively to the posterior; all except the first with wide diapophyses, which are also longer anteriorly (3d PSV has the longest transverse processes), decreasing in length progressively to the posterior, with chondrified tips (Figs. 6A and 6B). Diapophyses of vertebrae 2, 7, and 8 oriented anteriad, those of third and sixth—straight, and those of fourth and fifth oriented posteriad. Neural crests on PSV absent. Sacrum with slightly expanded diapophyses. Urostyle with weak dorsal crest running along about 75% of its shaft; ilia smooth, lacking dorsal crest (Fig. 6A).

Coracoids, scapulae, and suprascapulae present; first two fully ossified; suprascapula largely chondrified. Coracoids robust with narrow distal ends oriented anteriad; proximal ends greatly expanded (Fig. 6B). Omosternum absent. Procoracoid cartilage well-developed, extends from the scapula to the midline of the girdle. Clavicles present as slender, tiny, slightly recurved bones, lying on the procoracoid cartilage, not reaching scapula or the midline of the girdle (Fig. 6B). Sternum large, anterior portion consists of calcified cartilage (Fig. 6B), xiphisternum completely cartilaginous.

Bones of hands (Figs. 6C and 6D) with six largely calcified carpal elements: carpale distale I small, carpale distale II–IV fused into a single large element, prepollex and a large radiale lie between a small Y-element and an elongated ulnare. Metacarpals long and fully ossified; phalangeal formula: 2-2-3-3; all phalanges ossified; distal phalanx of finger I bobbin-shaped; terminal phalanges of fingers II–IV with greatly expanded, T-shaped tips, approximately three to four times wider than penultimate phalanges (Figs. 6C and 6D). Foot (Figs. 6E and 6F) with four tarsal elements, including ossified tarsale distale II–III, central and a prehallux; prehallux ossified. Metatarsals fully ossified, long and relatively more massive than metacarpals; phalangeal formula: 2-2-3-4-3; all phalanges ossified (Figs. 6E and 6F). Terminal phalanges of all toes with slightly expanded, T-shaped tips; equal in width to penultimate phalanges on toes I and V, slightly exceeding the width of penultimate phalanges on toes II–IV (Figs. 6E and 6F).

Larval morphology

Tadpole morphology description is based on a single larval specimen AUP-00509, Gosner stage 36, collected from the type locality. External appearance and coloration of the tadpole in life is shown in Fig. 8; morphological details in preservative are presented in Fig. 9. Body measurements are given below.

Figure 8 Tadpole of Siamophryne troglodytes Gen. et sp. nov. in life (AUP-00509; Gosner stage 36).

(A) In dorsal and (B) in ventral aspects. Scale bar equals to 5 mm. Photos by N. A. Poyarkov.

Figure 9 Tadpole of Siamophryne troglodytes Gen. et sp. nov. in preservative (AUP-00509; Gosner stage 36).

(A) In lateral, (B) in dorsal, and (C) in ventral views. Scale bar equals to 5 mm. Photos by T. Ruangsuwan.

The single encountered tadpole was attributed to S. troglodytes Gen. et sp. nov. based on the following evidence: (1) morphological features characteristic for Microhylidae larvae in general (flattened body, mouthparts lack lateral lobes, and keratinized structures); (2) collected in a crevice in the limestone cave, where adult animals were observed (see Figs. 10E and 10F); (3) species identification confirmed by mtDNA sequence of short 16S rRNA gene fragment (up to 500 bp), identical to that of adult specimens (GenBank accession number: MG682559, see Table 2).

Figure 10 Breeding habitat of Siamophryne troglodytes Gen. et sp. nov. at the type locality—Sai Yok District, Kanchanaburi Province, northern Tenasserim Region, western Thailand.

(A) Entrance to the limestone cave where the frogs were recorded; (B) female in situ sitting on the limestone wall of the cave; (C) male in situ sitting in a water-filled crevice; (D) female in situ on the wall of the cave (photos by M. Sumontha); (E, F) tadpole in situ in a water-filled crevice (photos by T. Ruangsuwan).

Standard tadpole measurements (AUP-00509, Gosner stage 36) (all in mm, taken by TR): ToL: 35.1; BL: 12.1; TaL: 21.8; BW: 7.9; BH: 5.4; TH: 4.0; SVL: 13.6; SSp: 5.6; UF: 1.0; LF: 1.0; IN: 1.2; IP: 1.7; RN: 1.9; NP: 1.9; ED: 1.0; MW: 2.6.

In dorsal view (Figs. 8A and 9B), body elongated (BW/BL 0.65) guitar-shaped with large bluntly rounded anterior part and a distinct narrowing behind the eyes that forms notably angular body edges; snout blunt with a shallow medial notch. From lateral view (Fig. 9A), both head and body strongly flattened. Tail strong, with well-developed muscles, less than two times longer than the body (Tal/BL 1.80), tail width at tail basis subequal to IOD; tail tip gently rounded. Tail fins very low, with even edges; dorsal fin not extending on the trunk, ventral fin same height as dorsal fin (LF/UF 0.94). A wide gular fold across the anterior half of body on the level of ca. 1/3 of BL (Figs. 8B and 9B). Spiracle medial, narrow, slit-like, transversal; located in the anterior half of body (SSp/SVL 0.41); covering membrane with even free margin. Vent tube medial, oblique.

Eyes dorsal, very small (ED/BL 0.08); with dorsolateral orientation of pupils. Small unpigmented narial protuberances present in nostril area. Mouth terminal, oriented anteroventrally, not visible from dorsal view (Fig. 9B), mouth opening transverse, slit-like from ventral view (Fig. 9C), relatively wide (MW/BW 0.33). From lateral view (Fig. 9A), upper labium slightly projecting and overhanging over the mouth opening; upper labium with even edge; lower labium short, with U-shaped edge. Protuberances in mouth angles, lateral lobes, and keratinized mouthparts absent. No papilla seen on mouth floor or mouth roof. Lingual anlage rounded.

Coloration

In life at stage 36 larva almost unpigmented and in situ appears translucent with dark-black colored eyes (Figs. 10E and 10F). In laboratory conditions, live tadpole (Fig. 8) shows semitransparent body with weak brownish pigmentation on dorsal surfaces of body and tail; a strongly-pigmented dark stripe runs along dorsal surface from interorbital region to tail base (Fig. 8A). Tail fins and ventral sides unpigmented, in life this allows to see heart, liver, and major blood vessels from ventral view (Fig. 8B). Limb buds pigmented with brown. In preservative, brown color fades to brown-gray (Fig. 9).

Natural history notes

Siamophryne troglodytes Gen. et sp. nov. has a troglophilous life style and to date is only known from a small limestone cave system in western Thailand. All specimens were collected within a narrow area inside a limestone cave located on elevation 440 m a.s.l. in a polydominant tropical forest in Sai Yok District, Kanchanaburi Province, western Thailand (Fig. 10A). The cave was examined twice on the 1st of August and the 27th of October, 2016. In both cases, adult specimens of S. troglodytes Gen. et sp. nov. were only recorded inside the cave, at a distance of more than 25 m from the entrance, sitting on walls of the cave (Figs. 10B and 10D) or hiding inside small caverns in limestone (Fig. 10C) or under flat stones. Despite the thorough search, no animals were recorded near the cave entrance or in the forest close to the cave. Animals were active from 23:00 to 24:00, when the air temperature inside the cave was 28 °C in August and 26 °C in October, in both cases with 100% humidity. No calling activity was recorded during both surveys. Diet and enemies of the new frog are unknown.

Three tadpoles (one of which was collected) were observed during the survey on the 1st of August, 2016, in a small water-filled cavity in the limestone on the floor of the cave, ca. 10 m from the cave entrance (Figs. 10E and 10F). The cavity was filled with water, the average depth was 4–5 cm; mosquito larvae (Chironomidae) were also observed in the same water body. Four other tadpoles (not collected) were discovered in another similar water-filled cavity inside the cave (30 m from the cave entrance).

The cave system where S. troglodytes Gen. et sp. nov. was discovered is inhabited by several species of bats which produce significant amount of guano that accumulates on the cave floor. According to a local guide, the locals mine this guano and that affects the ecosystem of the cave.

Comparisons

As for the genus. S. troglodytes Gen. et sp. nov. is a medium-sized frog (19 mm < SVL < 30 mm) and can be easily distinguished from all other microhylids inhabiting the mainland Southeast Asia by long limbs with digits bearing large disks, with those on fingers up to 2.5 times wider than the penultimate phalanges (vs. finger tips not expanded into prominent discs in Kalophrynus, Glyphoglossus, Microhyla, and Micryletta); by the absence of tibiotarsal projection and uniform dull-brownish coloration of ventral surfaces (vs. the presence of bony tibiotarsal projection and belly with bright saffron-yellow spots in Chaperina); by single small subarticular tubercle with indistinct borders at the basis of each finger (vs. subarticular tubercles of the fingers greatly enlarged to form accessory adhesive organs in Phrynella and Metaphrynella); by the lack of a bony ridge along the posterior border of each choana (vs. well-developed bony ridge along the posterior border of each choana in Kaloula), by the presence of a distinct tympanum (vs. hidden tympanum in genera Glyphoglossus, Microhyla, Micryletta, Kaloula, Phrynella, and Metaphrynella).

The new species can be further distinguished from Gastrophrynoides, another Asterophryinae genus inhabiting the mainland Southeast Asia, by slender body habitus (vs. body habitus robust in Gastrophrynoides, see Fig. 2 for details), by a short rounded snout, that is 0.8–1.0 times the ED (vs. long, pointed snout that is 2.6–3.0 times the diameter of the eye in Gastrophrynoides) and by a well-developed tympanum (vs. tympanum hidden in Gastrophrynoides).

Distribution

As for the genus. At present, S. troglodytes Gen. et sp. nov. is known from a single limestone karst cave in Sai Yok District of Kanchanaburi Province in western Thailand. To date, numerous surveys in the nearby karst massifs have not yielded discoveries of additional populations of the new species. However, further fieldwork in Kanchanaburi Province of Thailand and the adjacent parts of Tanintharyi Division of Myanmar are required.

Conservation status

This species was unknown to science and even to local people for a surprisingly long time despite intense human activity in this region; that suggests a possibly very limited range. Despite our numerous attempts to find additional localities of the new species in adjacent limestone areas, we failed to detect other populations of this troglophilous frog. In this regard, we consider a disclosure of the detailed collecting locality information premature. In the case of narrow-ranged species of amphibians, the locality information should be released only after effective conservation measures have been taken and legal protection status have been established (see discussion in Hou et al., 2014). Until then, this information can be requested from the School of Agriculture and Natural Resources, University of Phayao. Currently, the only known habitat of S. troglodytes Gen. et sp. nov. is endangered (EN) due to illegal guano mining with the use of explosives, which is destroying the cave ecosystem. Given the available information, we suggest S. troglodytes Gen. et sp. nov. to be considered as an EN species following IUCN’s Red List categories (IUCN Standards and Petitions Subcommittee, 2016).

Etymology

The specific name “troglodytes” is a Latin adjective in the nominative singular meaning “cave-dweller”, derived from the Greek “τρωγλoδύτης”, with “trogle” meaning “hole, mouse-hole” and “dyein” meaning “go in, dive in”; referring to the troglophilous biology of the new species, which was recorded only in a limestone karst cave system.

Suggested common names

We recommend the following common names for the new species: “Tenasserim Cave Frog” (English); “Eung Tham Tenasserim” (Thai).

Discussion

Systematics and biogeography

In the present paper, we report a new lineage of Asterophryinae microhylid frogs from western Thailand, Tenasserim region of mainland Southeast Asia. As predicted by Kurabayashi et al. (2011), S. troglodytes represents an ancient lineage of Asterophryinae frogs distributed deep in the mainland Southeast Asia (Indo-Burma); it is reconstructed as a sister lineage to Gastrophrynoides—the only asterophryine found in the areas derived from the Eurasian landmass (Sundaland) until now. Our discovery of S. troglodytes has a significant biogeographic importance as, for the first time, it documents the presence of Asterophryinae lineage north of the Isthmus of Kra (see Fig. 1) and thus strongly supports the Asian dispersal scenario for the group, suggested by Kurabayashi et al. (2011). According to this scenario, the splitting of the lineage that gave birth to the ancestors of Asterophryinae, Microhylinae, and Dyscophinae occurred in the Indian landmass during the late Cretaceous (around 70 Ma). Following the collision of the India plate and the Eurasia during the Eocene (around 48 Ma, Kurabayashi et al., 2011), Asterophryinae common ancestor would have colonized mainland Southeast Asia and split into Gastrophrynoides + Siamophryne lineage and the ancestor of the “core” Asterophryinae 1 lineage. During the late Oligocene (around 25 Ma) the “core” Asterophryinae 1 ancestorsfurther dispersed eastwards from mainland Asia to Australasian landmass via islands and/or short sea straits, where they undergone a comparatively recent and fast adaptive radiation following the orogenetic processes in the Fold Belt and the Australian Craton of New Guinea (Rivera et al., 2017) and subsequently colonized northern Australia and adjacent smaller islands.

Thus, our work demonstrates that S. troglodytes represents an old lineage of initial radiation of Asterophryinae sensu lato, which took place in the mainland Southeast Asia. The discovery of S. troglodytes goes in line with the biogeographic pattern reported for the Australasian frog family Ceratobatrachidae (Natatanura) by Yan et al. (2016): until recently, this family was only known from the South-West Pacific to the island archipelagos of South Asia, with primary centers of species diversity in Philippines and Solomon-Bismarck Archipelago. However, a recent phylogenetic study greatly extended the westernmost border of geographic distribution of the primarily Australasian family Ceratobatrachidae into western Thailand (Tenasserim) and Himalaya and assigned early branching events in this family to the lineages that are now exclusively represented by the mainland species with ranges restricted to Himalaya and Tibet (Yan et al., 2016). Hence, the biogeographic scenarios for at least two most speciose frog families of the Australasia suggest that their origin and early cladogenesis in the mainland Southeast Asia was followed by dispersal into Australasian archipelago and subsequent radiation. Our study encourages further field research and phylogenetic studies on Southeast Asian frogs, since they may yield discoveries of new taxa and unexpected biogeographic patterns.

Natural history and reproductive biology

Limestone caves provide unique combinations of ecological features, such as rocky vertical substrates, climatic stability, low illumination, relaxed predation, and reduced prey base. Cave-dwelling taxa of amphibians and reptiles are often characterized by unique morphological adaptations, which are supposed to increase locomotion efficiency on vertical and inverted surfaces within cave-like microhabitats. In cave-dwelling ecomorphs of Cyrtodactylus geckoes these often include comparatively shorter trunks; longer and thinner limbs; flatter, narrower, and sometimes longer heads; and relatively larger eyes (Grismer & Grismer, 2017). S. troglodytes, as compared to its sister genus Gastrophrynoides, shows a number of morphological differences, which might be connected to its troglophilous life style. These include slender body habitus with notably longer limbs (see Fig. 2 for comparison), long and thin fingers and toes with tips expanded to large discs (see Fig. 5), slightly longer head (HL/SVL ratio 0.31 (0.30–0.33; N = 11) in S. troglodytes vs. 0.28 (0.27–0.29; N = 3) in Gastrophrynoides immaculatus), and comparatively larger eyes (EL/HL ratio 0.41 (0.37–0.46; N = 11) in S. troglodytes vs. 0.20 (0.19–0.20; N = 3) in G. immaculatus). Higher distances between the opposing hands and opposing feet with enlarged digit disks and larger eyes might facilitate locomotion on vertical and inverted walls of limestone caves in low illumination conditions.

The life cycle of many Asterophryinae species is still unknown and females of most species are rare in museum collections. For the Australo-Papuan “core” Asterophryinae lineage there is sufficient evidence that allows to assume that all of the members of that lineage have direct development—a life cycle with metamorphosis taking place within the egg (Menzies, 2006; Günther et al., 2012); this contrasts sharply with the Southeast Asian Microhylinae and Kalophryninae, all of which have free-living tadpoles. The reproductive biology and development of Gastrophrynoides is completely unknown (Parker, 1934; Chan et al., 2009); to our knowledge, no information on female or egg morphology in this genus is available either. Our work shows that S. troglodytes possesses a life cycle with a free-living larval stage; tadpole morphology of the new genus is very peculiar and was not reported for any other microhylid so far. This is the first report of a free-living tadpole for Asterophryinae, as well as the first known Asterophryinae with a troglophilous life style. Our description is based on a single larval specimen, further studies on the reproductive biology and development of S. troglodytes as well as of the genus Gastrophrynoides are required to understand the details of life cycle in this lineage of Asterophryinae. It is assumed that the divergence between the “core” Asterophryinae lineage and the Siamophryne + Gastrophrynoides lineage took place no later than the Eocene (Kurabayashi et al., 2011). Further research might reveal more significant differences in morphology and natural history of these lineages.

Conservation

Our study adds a new genus of frogs to the batrachofauna of Thailand and Indochina; according to our knowledge, S. troglodytes is endemic to a small limestone cave system in Sai Yok District of Kanchanaburi Province in western Thailand. This small area is known for an exceptionally high number of endemic species of squamates discovered by previous herpetofaunal surveys, including five endemic gecko species and two endemic species of snakes (see Sumontha et al., 2017). The reasons behind such exceptional herpetofaunal endemism are yet unclear; it is remarkable that most reptiles that are endemic to Sai Yok District are associated with limestone habitats; this is also the case for S. troglodytes. Like the other representatives of Thai endemic herpetofauna that live in caves or their direct surroundings, the newly discovered genus has to cope with a high degree of human disturbance (Pauwels, Sumontha & Ellis, 2016), in this case—due to illegal mining of bat guano that causes destruction and modification of the cave ecosystem. This may pose a serious threat for S. troglodytes in the future; immediate assessment of its conservation status on national and international levels, conservation measures at the type locality that will minimize the modification of the natural habitat by humans, as well as intensive surveys for the new localities are essential for the protection of this relict frog lineage.

The discovery of S. troglodytes further demonstrates the key role of limestone karst areas as arks of highly endangered biodiversity. Geological structure, erosion, and subterranean water drainages of limestone areas permanently provide unique diversity of microrefugia with numerous shelters, like cracks and caves (Clements et al., 2006). These humid microhabitats might provide an efficient environmental buffer for small vertebrates during periods of climate change (Glaw, Hoegg & Vences, 2006). Isolated limestone massifs throughout the world are known as hotspots of vertebrate endemism and persistence (Oliver et al., 2017a). Karst areas act as microrefugia for relict amphibian species (see Sket, 1997; Min et al., 2005; Glaw, Hoegg & Vences, 2006; Milto et al., 2013). Karsts are also known as important “biodiversity arks” for both surface and cave faunas (Clements et al., 2006) with numerous new species of amphibians and reptiles being discovered from limestone areas (e.g., see Köhler et al., 2010; Nazarov et al., 2014; Rakotoarison et al., 2017; Grismer & Grismer, 2017; Grismer et al., 2017, and references therein). Ironically, though acting as major biodiversity hotspots, limestone karst areas are critically EN due to intensive deforestation and cement manufacturing; their continued exploitation for limestone cannot be stopped (Clements et al., 2006). Our study thus calls for further focused survey and conservation efforts on karst herpetofauna in Southeast Asia.

Conclusion

Siamophryne troglodytes, a new genus and species of microhylid frogs from western Thailand, belongs to the subfamily Asterophryinae, which is most diverse in Australasia. Siamophryne and its sister genus Gastrophrynoides are the only two asterophryine lineages found in the areas derived from the Eurasian landmass. Our work demonstrates that S. troglodytes represents an old lineage of the initial radiation of Asterophryinae which took place in the mainland Southeast Asia. Our results strongly support the “out of Indo-Eurasia” biogeographic scenario for this group of frogs. To date, the new frog is the only known asterophryine with a free-living tadpole and troglophilous life style. Further studies might reveal new members of Asterophryinae in the mainland Southeast Asia.

Supplemental Information

Supplemental Information 1 The final alignment of the obtained sequences of the 16S rRNA–12S rRNA fragment of Siamophryne troglodytes Gen. et sp. nov. specimens, as well as Microhylidae and non-Microhylidae outgroup taxa, subjected to the phylogenetic analyses.

Alignment in fasta (.fas) file format.

Click here for additional data file.

We would like to thank the Laboratory Animal Research Center, University of Phayao and The Institute of Animal for Scientific Purposes Development Thailand for the permission to work in the field and Nirut Chomngam, Chaowalit Songsangchote, and Akrachai Aksornneam for help during the field work. We are most grateful to Yu Lee (Chinese Culture University, Taipei) for providing photos of Gastrophrynoides. NAP thanks Valentina D. Kretova (Biological Faculty, Lomonosov Moscow State University) for help with preparation of figures, Vladislav Gorin (Biological Faculty, Lomonosov Moscow State University) for help and assistance in the lab, Duong Van Tang (Biological Faculty, Lomonosov Moscow State University) for help with phylogenetic analyses and Alexandra A. Elbakyan for help with accessing required literature. We are indebted to Evgeniya N. Solovyeva (Zoological Museum of Moscow University) for help with primer design and to Egill Scallagrimsson and Natalia Ershova for proofreading. We express our sincere gratitude to Gabriela Parra Olea and the three anonymous reviewers for their useful suggestions on the earlier version of the manuscript.

Additional Information and Declarations

Competing Interests

Author Contributions

Animal Ethics

Field Study Permissions

DNA Deposition

Data Availability

New Species Registration

The authors declare that they have no competing interests.

Chatmongkon Suwannapoom conceived and designed the experiments, performed the experiments, analyzed the data, contributed reagents/materials/analysis tools, prepared figures and/or tables, authored or reviewed drafts of the paper, approved the final draft.

Montri Sumontha analyzed the data, contributed reagents/materials/analysis tools, authored or reviewed drafts of the paper, approved the final draft.

Jitthep Tunprasert analyzed the data, authored or reviewed drafts of the paper, approved the final draft.

Thiti Ruangsuwan performed the experiments, analyzed the data, contributed reagents/materials/analysis tools, authored or reviewed drafts of the paper, approved the final draft.

Parinya Pawangkhanant analyzed the data, contributed reagents/materials/analysis tools, authored or reviewed drafts of the paper, approved the final draft.

Dmitriy V. Korost performed the experiments, analyzed the data, contributed reagents/materials/analysis tools, authored or reviewed drafts of the paper, approved the final draft.

Nikolay A. Poyarkov conceived and designed the experiments, performed the experiments, analyzed the data, contributed reagents/materials/analysis tools, prepared figures and/or tables, authored or reviewed drafts of the paper.

The following information was supplied relating to ethical approvals (i.e., approving body and any reference numbers):

Specimens collection protocols were approved by the Institutional Ethical Committee of Animal Experimentation of the University of Phayao (certificate number UP-AE59-01-04-0022 issued to Chatmongkon Suwannapoom) and strictly complied with the ethical conditions by the Thailand Animal Welfare Act.

The following information was supplied relating to field study approvals (i.e., approving body and any reference numbers):

Specimens were collected and exported with permission of the Animal Research Centre, University of Phayao, Thailand, and of the Institute of Animals for Scientific Purpose Development, Bangkok, Thailand, under the jurisdiction of the National Research Council of Thailand, permit no. U1-01205-2558, issued to Chatmongkon Suwannapoom.

The following information was supplied regarding the deposition of DNA sequences:

The obtained sequences described here are accessible via GenBank under the accession numbers MG682553 to MG682559.

The following information was supplied regarding data availability:

Specimens and tissues examined in the present manuscript are stored in the herpetological collections of the School of Agriculture and Natural Resources, University of Phayao (AUP, Phayao, Thailand) and of the Zoological Museum of Moscow University (ZMMU, Moscow, Russia).

The following information was supplied regarding the registration of a newly described species:

Genus name: Siamophryne

urn:lsid:zoobank.org:act:CD1B3316-201B-488C-B679-02A1DA47CE5C

Species name: Siamophryne troglodytes

urn:lsid:zoobank.org:act:639A70F2-4743-47D5-8E35-D8F4D3D51123

Publication LSID:

urn:lsid:zoobank.org:pub:C8BD1C1D-0553-4662-8DE5-4337CD69E3B9

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
