# Peer review of "A striking new genus and species of cave-dwelling frog (Amphibia: Anura: Microhylidae: Asterophryinae) from Thailand"

_PeerJ, doi:10.7717/peerj.4422_

## Round 0.1 · original submission · Minor Revisions

This paper describes a new genus and species of microhylid frog from Thailand. The article is well written, the results are clearly presented, and this manuscript represents an important contribution to the knowledge of Herpetofauna from southeast Asia. However, I agree with the reviewers in that, the discussion is too long, and discusses subjects deeply discussed in other papers that add little to the recognition of Siamophryne troglodytes. I would suggest to the authors to pay close attention to all of the reviewers comments and to follow their advice especially in terms of cleaning and shortening the discussion.

Reviewer 1 ·

Basic reporting

The English is clear and the style concise, in general the manuscript is easily read and perfectly structured. Literature review complete and up to date.

I would recommend to subdivide the discussion, for example systematics and biogeography, evolution of reproductive biology and conservation.

The illustration of the paper is outstanding.

The authors provide a fasta alignment with the new sequences. In this case, using ribosomal genes, I would suggest to include the whole alignment to facilitate replicability of analyses.

Experimental design

The description of the new species is presented in a more general systematic and biogeographic scenario, highlighting the interest of the finding. The manuscript fulfills the research standards in the field.

Validity of the findings

The manuscript presents not only the finding of a new genus and species, but provides an important contribution to the knowledge on the biogeography of the region.

The discussion is based on solid results. Even if phylogenetic relationships are not resolved in every clade, the focus is set on a well supported part of the tree.

Additional comments

The authors have done a great job in converting the description of a new genus and species into a notable contribution to the evolution, systematics and biogeography of the Microhylidae. I think the manuscript is suitable for publication and have only a few minor comments and suggestions.

I think the most important part of the discussion is the biogeographic history of the subfamily. They provide a detailed scenario. However, I think it would be strengthened if the authors test it with estimates of times of divergence of the different lineages of the group. Perhaps some of the calibration points used in previous papers could be used. Regarding phylogenetic analyses it would be interesting to check for optimal partition schemes for the data.

In the introduction, when writing about subfamilies (lines 9-10), indicate the degree of coherence between morphological and molecular classifications.

I think the introduction should be closed highlighting the implications of the new species in the biogeography and evolution of the group, as these are the main topics in the discussion.

You sometimes write the new name "Siamophryne troglodytes Gen. et sp." (example line 668) and other times "Siamophryne troglodytes Gen. et sp. nov." (example lines 687-688). Please check and unify. Is it necessary to add " Gen. et sp. nov." every time you use the name after the description?

Lines 90-92, indicate the articles of the code supporting this point.
Line 122, a bit confusing. Did you use different primers for PCR and sequencing? If so indicate which ones for which process.

Lines 128-129, confusing. Did you use all the PCR product for electrophoresis? I suggest "PCR products were visualized using 1.5% agarose electrophoresis in presence of ethidium bromide".

Line 141, analyses

Line 142, "sequences from other"

Lines 150-170, check this paragraph, it seems that some genus names are missing their authors. These part is a bit messy, perhaps it would be better to reduce it and refer to Table 2.

Line 314, "Austrochaperina is paraphyletic...". Is it possible that they are misidentifications?

Line 320-321, I think the sentence should be rewritten: p-distances among and within gene fragments or among and within genera?

Line 328-329, "gene of the new species (Table 3)".

Line 348, minimum size should be 19 mm, not 20.

Line 374-375, " Asterophryinae genera which have symphignathine".

Line 423-424, "expanded and broad cultiform process of parasphenoid in..."

Line 457, I think numbers are provided for only 9 specimens.

Line 547, " are larger than males".

Line 793-794, just before this sentence you stated that nothing is known about the reproductive biology of Gastrophrynoides, so it sounds a bit speculative.

In table 2 and its caption, would not it be possible to use the full name of the new species? Same for table 3.

In table 3 caption it says: " The mean uncorrected p-distances for the ingroup are shown in the diagonal and shaded with grey.". Are these not distances within some genera? Check.

Reviewer 2 ·

Basic reporting

This paper describes a very interesting new species of microhylid frog from Thailand, Generally the quality of the research and inferences made based on the data seem highly justified and reasonable.

This said, there is perhaps a missing opportunity to link in with other similar finds of high diversity, restricted or relict diversity in karst habitat, e.g see a number of papers on herps:
Glaw et al. 2006 Zootaxa, 1334, 27-43 Tsingymantis
Kohler et al. 2010 Journal of Zoology 282, 21-38.
Oliver et al. 2017 Diversity and Distributions, 23, 53-66
Grismer et al. 2017 eg latest in Biol. J Linn Soc (and many others)
And perhaps Clements et al. 2006, Bioscience 53,733-746 - for a more general ref

Likewise in lizards a least there is literature on morphological adaptations that characterise cave dwelling taxa, does this new species so any adaptitions typical of cave vertebrates – reduced pigmentations, longer limbs etc?

The discussion of biogeography seems a little long, it already pretty well established that Asterophryne has Asian origins, this new find supports and strengthens this interpretation, but less discussion of the alternative hypothesis (Gondwanan origins) is needed.

Results – suggest delete all discussion of relationships within the Melanesian Asterophyrninae (lines 302-314)– it is not relevant to this paper, the data upon which this discussion is based is not new, and more recent studies – e.g Rivera et al 2017 have better data for this job anyway

Experimental design

All fine.

Validity of the findings

Data, inference and overall findings seem robust.

Conclusions are all reasonable, however as noted above there are several paragraphs focused on relationships within the New Guinea Asterophyrnes that are recapitulations of data presented elsewhere, the data avaliable (12S and 16S) does a poor job, and better phylogenies have already been published. These sections should be cut.

As noted above, most speculation is reasonable, but I would also suggest that is a missed oppurtunity here to link into and emerging literature on a global story of karst herpetofaunas (endemism, relictualism and specialisation).

Additional comments

Minor comments

Line 17 – Gondwanan landmass = Gondwana?

Lines 35-45 While the placement of Gastrophrynoides in association with the Melanesian taxa is probably right, its placement within that clade is likely wrong (based as it was on small amout of heavily saturated mtDNA), and as alluded to later, so why report this result in so much detail?

Lines 49-51 – saying same thing twice, but in different ways

Line 72 – personal bugbear, but, please use decimal co-ordinates – the more people use these the easier processing this data will be

Line 118 – could add a few more taxonomic studies – e.g.
Hoskin 2004 Australian Journal of Zoology, 52, 237-269
Oliver et al. 2017 PeerJ 5,e3077
Conversely Blackburn paper was not focused on Asterophryninae

Line 150-162 – large bracketed section is very awkward, would easirer to read if broken up into several sentences

Line 262 – sequence alignment, how was this done, and presumably there were unalignable regions that were chopped out?
Line 374 – ‘thereunder’ = “thereby?”

Line 388 – s. lato = sensu lato (spell out fully, or abbreviate full)

Line 396 “can be distinguished from the new genus by reduced clavicles (vs. clavicles present)” – this does not make sense

Line 468 – ‘limbs very long’ – would be great to add some relative proportions measures to expand on this

Line 479-480 – this is not a trait that can be scored from specimens, so I would suggest removing

Line 486 – change “’,” to “;”

Line 496 – “more than two times shorter” = “less than half length”?

Line 537 – what colour is the ‘dark mottling”?

Line 668 – Comparisons (or somewhere) – if possible it would be nice to include an image of Gastrophrynoides to emphasis how different these taxa are, even though they are sister species. They provide another example of the ecological plasticity of microhylids – e.g Blackburn kaloula etc.

Line 687 – are there many similar caves nearby? Likewise is there exposed karst habitats that they could live amongst?

Line 752-776 – this paragraph restates other peoples results and can be deleted

Line 777-778 - delete first sentence – no one has ever found a microhylid tapdole in New guinea – they are all almost certainly all direct developers

Final para of discussion – see above major comments about a missing opportunity to link in with a broader literature on cave endemism in herps, and the conservation importance of caves

Line 821 – change to ‘only two’

Reviewer 3 ·

Basic reporting

The article is well written and results are clearly presented with the help of tables and figures that have been carefully prepared.

Experimental design

The article reports the finding of a highly divergent lineage of microhylid frog, which carries important biogeographic implications. The authors show the distinctiveness of this new lineage with mitochondrial DNA sequences, morphological (internal and external), and ecological data. The phylogenetic analyses are correctly conducted and presented and the description of the osteology is very comprehensive and informative.

Validity of the findings

While only mtDNA sequences are used to reconstruct phylogenetic affinities of the new taxon, the results clearly show its position as an early branching member of the Asterophryinae. The biogeographical implications of the results of these analyses are carefully discussed in light of previous hypotheses.

Additional comments

I congratulate the authors on a very thorough study on a very interesting finding. I only have a few minor suggestions for corrections in the text (see below):

Line 18: Madagascar

Line 146: in fact, it's 56 microhylids plus the outgroup

Line 164: Melanobatrachus

Line 181: Confidence in tree topology was evaluated/assessed rather than tested

LIne 223: add abbreviation (2-5 TOEL)

Line 230: ...those of Savage

Line 290: Asterophryinae consists of two major...

Line 293: this should read east of the Wallace line, not west

Line 317: Gastrophrynoides

Line 328: this should be 12S rRNA-16S rRNA

Line 337: inhabits

Line 412: lacks these structures

Lines 421 and 423: cultriform process

Line 534: refer to Fig. 5

Line 614: this would be Table 2, not Table 1

Line 717: Gastrophrynoides

Lines 727, 730, 731: rather than "had", I'd either use "would have" or delete "had" altogether in the three cases

Line 748: replace "initial origination and cladogenesis" by "origin and early cladogenesis"

Line 768: Liophryne

Line 776: replace "better" for "robust"

Line 785: replace "as well" for "either"

Line 814: relict

Line 820: Gastrophrynoides (check spelling throughout)

Figure 2 legend: GB accession numbers are listed in Table 2, not Table 1. Also, either explain that the A in the tree corresponds to the Asterophryinae node or delete from the figure

Figure 6 legend: check spelling for phalanges (mis-spelled twice). Also, although it is referred to in the legend, I could not find the prehallux in the figure

Figure 7 legend: although it is referred to in the legend, I could not find the vomer in the figure

Table 1: there are six primers listed to amplify the 12S fragment, perhaps you could clarify in the text whether these correspond to nested PCRs or some are sequencing primers. If this information is presented in Nguyen et al (in press) forget about this

---

## Round 0.2 · accepted · Accept

I have read the revised version of: A striking new genus and species of cave-dwelling frog (Amphibia: Anura: Microhylidae: Asterophryinae) from Thailand and I acknowledge the effort made by the authors with regards to the reviewer’s comments. I am happy to accept this manuscript for publication.